



# Smoothed Particle Hydrodynamics Implementation of the Standard Viscous-Plastic Sea-Ice Model and Validation in Simple Idealized Experiments

Oreste Marquis[1], Bruno Tremblay[1], Jean-François Lemieux[2], and Mohammed Islam[3]

[1]Department of Atmospheric and Oceanic Sciences, McGill University, Montréal, Québec, Canada

[2]Environnement et Changement Climatique Canada, Dorval, Québec, Canada

[3]The Ocean Coastal and River Engineering Research Centre of the National Research Council, Canada

**Correspondence:** Oreste Marquis (oreste.marquis@mail.mcgill.ca)

**Abstract.** The Viscous-Plastic (VP) rheology with an elliptical yield curve and normal flow rule is implemented in a Lagrangian modelling framework using the Smoothed Particle Hydrodynamics (SPH) meshfree method. Results show, from perturbation analysis of SPH sea-ice dynamic equations, that the classical SPH particle density formulation expressed as a function of sea-ice concentration and mean ice thickness, leads to incorrect plastic wave speed. We propose a new formulation for particle density that gives a plastic wave speed in line with theory. In all cases, the plastic wave in the SPH framework is dispersive and depends on the smoothing length (i.e., the spatial resolution) and on the SPH kernel employed in contrast with its finite difference method (FDM) implementation counterpart. The steady-state solution for the simple 1D ridging experiment is in agreement with the analytical solution within an error of 1%. SPH is also able to simulate a stable upstream ice arch in an idealized domain representing the Nares Strait in low wind regime (5.3 $[\mathrm{m} \cdot \mathrm{s}^{-1}]$) with an ellipse aspect ratio of 2, an average thickness of 1 $[\mathrm{m}]$ and free-slip boundary conditions in opposition to the FDM implementation that requires higher shear strength to simulate it. In higher wind regime (7.5 $[\mathrm{m} \cdot \mathrm{s}^{-1}]$) no stable ice arches are simulated — unless the thickness is increased — and the ice arch formation showed no dependence on the size of particles contrary to what is observed in the discrete element framework. Finally, the SPH framework is explicit, can take full advantage of parallel processing capabilities and show potential for pan-arctic climate simulations.



## 1 Introduction

Sea-ice is an important component of the Earth's system to consider for accurate climate projection. Numerical models for geophysical sea-ice have historically employed a continuum approach where the material is discretized on an Eulerian mesh using various constitutive relations. For example, the standard Viscous-Plastic model (Hibler, 1979) or modifications (e.g., Elastic-Viscous-Plastic or EVP and Elastic-Plastic-Anisotropic or EPA; Hunke and Dukowicz, 1997; Tsamados et al., 2013),
solves a set of partial differential equations using the finite-difference method (FDM). FDM is the simplest method to discretize and solve partial differential equations numerically. However, it is based on a local Taylor series expansion to approximate the continuum equations and construct a topologically rectangular network of relations between nodes (e.g., Arakawa grids).

Even though the VP (and EVP) rheologies are commonly used to describe sea-ice dynamics and are able to capture important large-scale deformation features (Bouchat et al., 2022; Hutter et al., 2022), they still have difficulties to represent smaller scale
properties (Schulson, 2004; Weiss et al., 2007; Coon et al., 2007) such as Linear Kinematic Features (LKFs) unless run at very high resolution ($\approx 2\,\mathrm{km}$, Ringeisen et al., 2019; Hutter et al., 2022). To improve the simulation of small-scale ice features and to alleviate the problem of FDM with complex geometries (Peiró and Sherwin, 2005), the community also considered new sea-ice rheologies (Schreyer et al., 2006; Girard et al., 2011; Dansereau et al., 2016; Ringeisen et al., 2019) and explored different space discretization frameworks like the finite-element method (FEM) (Rampal et al., 2016; Mehlmann et al., 2021),
the finite-volume method (FVM) (Losch et al., 2010; Adcroft et al., 2019) or the discrete-element method (DEM) (Hopkins and Thorndike, 2006; Herman, 2016; Damsgaard et al., 2018).

In recent decades, spatial resolution of sea-ice models became comparable to the characteristic length of the ice floes. This makes the continuum assumption of current FDM, FVM and FEM models questionable. Also, Eulerian models are known to have difficulties determining the precise locations of inhomogeneity, free surfaces, deformable boundaries and moving
interfaces (Liu and Liu, 2010). These shortcomings have led to an increase interest in the DEM approach. Another advantage of using DEM is that granularity of the material (Overland et al., 1998) is directly represented using discrete rigid bodies from which the physical interactions are calculated explicitly in the hope that large-scale properties naturally emerge. In practice, the emergent properties still depend on the assumed floe size and the nature of collisions. Nevertheless, DEM easily captures formation of cracks, leads and large deformation making it a consistent framework for the numerical simulation of granular
material like sea-ice (Fleissner et al., 2007).

Despite the shortcomings of the continuum approaches, FDM, FVM and FEM are still the most commonly used framework in the community because they have been developed and tested for a longer period, they are well understood, more computationally efficient and easily coupled for synoptic scale simulations. In an attempt to take advantages of both continuum and discrete formulation, blends between the two approaches have been proposed — e.g., the finite-discrete element (Lilja et al.,
2021) or the material-point method (Sulsky et al., 2007). Those framework, however, still use a mesh to solve the dynamic equations in addition to considering sea ice as discrete elements making them even more computationally expensive. Finally, a fairly new approach for sea-ice modelling — also taking from both continuum and discrete framework — uses a Lagrangian meshfree continuous method called smoothed particle hydrodynamics (SPH) (Lucy, 1977; Gingold and Monaghan, 1977).



This meshfree method is known to facilitates the numerical treatment and description of free surfaces (Liu and Liu, 2010)
which are common in sea-ice dynamics with polynyas, LKFs, free drifting ice floes and unbounded ice extent. As in DEM, the
physical quantities are carried out by particles in space (an analogy for ice floes in the real world), but evolve according to the
same dynamic equations used in the continuum approach. Furthermore, the method has the advantage of treating the system of
equations in a Lagrangian framework discretized explicitly making it well suited for parallelization.

SPH has been used successfully for the modelling of other granular materials such as sand, gravels and soils (Salehizadeh
and Shafiei, 2019; Yang et al., 2020; Sheikh et al., 2020). In the context of mesoscale and larger sea-ice modelling, Gutfraind
and Savage (1997) initiated the SPH study of sea-ice dynamics using a VP rheology based on a Mohr-Coulomb failure criterion.
The ice concentration and thickness were fixed at 100% and 1 [m] with a continuity equation expressed in terms of a particle
density. The internal ice strength between particle was derived diagnostically from ice density assuming ice was a nearly
incompressible material. Later, Wang et al. (1998) developed a sea ice model of the Bohai Sea (east coast of China) using
an SPH viscous-plastic rheology (Hibler, 1979) with continuity equations for ice concentration and mean thickness, and ice
strength calculated from static ice jam theory (Shen et al., 1990). Following Wang et al. (1998), Ji et al. (2005) implemented
a new viscoelastic-plastic rheology that was in better agreement with observations from the Bohai Sea. Recently, Staroszczyk
(2017) proposed a sea ice model considering ice to behave as a compressible non-linear viscous material with a (dimensionless)
contact length dependent parameterization for floe collisions and rafting (Gray and Morland, 1994; Morland and Staroszczyk,
1998). In all of the above, except for Gutfraind and Savage (1997), the same ice particle density definition is used.

In this work, we use the standard VP sea-ice model with an elliptical yield curve and normal flow rule (Hibler, 1979), and
propose a reformulation of the ice particle density that is internally consistent with the model physics. One goal of the study
is to investigate differences coming from the numerical framework. To this end, we theoretically investigate the plastic wave
propagation, a fundamental property of a sea-ice physical model, throughout a 1D perturbations analysis and we test the model
in a ridging and ice arch experiment following earlier works by Lipscomb et al. (2007); Dumont et al. (2009); Rabatel et al.
(2015); Dansereau et al. (2017); Williams et al. (2017); Damsgaard et al. (2018); Ranta et al. (2018); Plante et al. (2020);
West et al. (2022). We chose to investigate the SPH method performance on a ridging experiment since it has an analytical
steady-state solution that can be used to establish the model accuracy and it is possible to evaluate the coherent evolution of
the continuity equations. We also test SPH performance on ice arches simulation since this classic problem is an example
of large-scale features resulting from small-scale interaction involving fractures of the material. The two experiments allow
a direct comparison with previous work and identify advantages and disadvantages with the continuum and discrete sea-ice
dynamic. The long-term goal is to lay the foundation for an SPH sea-ice formulation that can be used in synoptic scale models.

The paper is organized as follows. In section 2, the SPH framework and how the sea-ice VP rheology, momentum and
continuity equations can be implemented in this framework are described. Results of a plastic wave propagation analysis,
ridging experiments, and ice-arching simulations are presented in the section 3. Finally, section 4 discuss the SPH advantages
and limitations of the framework and model developed and concludes.





## 2 Model

### 2.1 Smoothed Particle Hydrodynamics (SPH)

The SPH method is at the interface between finite element method and discrete element methods. In this framework any
function $f(\boldsymbol{r})$ at a point $\boldsymbol{r}$ is approximated from neighbouring values in the parameter space $f(\boldsymbol{r}')$ using an integral interpolant
(see figure 1) :

$$f(\boldsymbol{r}) = \int_{\mathcal{V}} f(\boldsymbol{r}')W(|\boldsymbol{r} - \boldsymbol{r}'|, l)d\boldsymbol{r}', \tag{1}$$

where $W(|\boldsymbol{r} - \boldsymbol{r}'|, l)$ is the interpolating kernel and $\mathcal{V}$ is the entire space volume. In two dimensions, the space volume is an
area $\mathcal{A}$ and the kernel has units of $[\mathrm{m}^{-2}]$. This integral interpolant approximation is based on the singular integral mathematical
framework of Natanson (1961) and imposes the following restrictions on the kernel:

$$\int_{\mathcal{A}} W(|\boldsymbol{r} - \boldsymbol{r}'|, l)d\boldsymbol{r}' = 1, \tag{2}$$

and

$$\lim_{l \to 0} W(|\boldsymbol{r} - \boldsymbol{r}'|, l) = \delta(\boldsymbol{r} - \boldsymbol{r}'), \tag{3}$$

where $l$ is the smoothing length of the kernel and $\delta$ is the Dirac delta function. Using the particle approximation, Eq. (1) can
be written as a weighted summation over all neighbouring points within the area $\mathcal{A}$:

$$f(\boldsymbol{r}_p) \approx \sum_{q=1}^{N} f(\boldsymbol{r}_q)W(|\boldsymbol{r}_p - \boldsymbol{r}_q|, l_p)\Delta\mathcal{A}_q \approx \sum_{q=1}^{N} f(\boldsymbol{r}_q)W(|\boldsymbol{r}_p - \boldsymbol{r}_q|, l_p)\frac{m_q}{\rho_q}, \tag{4}$$

where $N$ is the number of points in space referred as neighbour particles, $\Delta\mathcal{A}_q (= m/\rho)$ is the area associated with the particle
$q$, $m$ represent the mass [kg] and $\rho$ is the 2D density $[\mathrm{kg} \cdot \mathrm{m}^{-2}]$. From the above approximations, we reformulate differential
operators relevant to our study in their discrete SPH forms. We write the divergence of a vector field ($\boldsymbol{V}$), the divergence of a





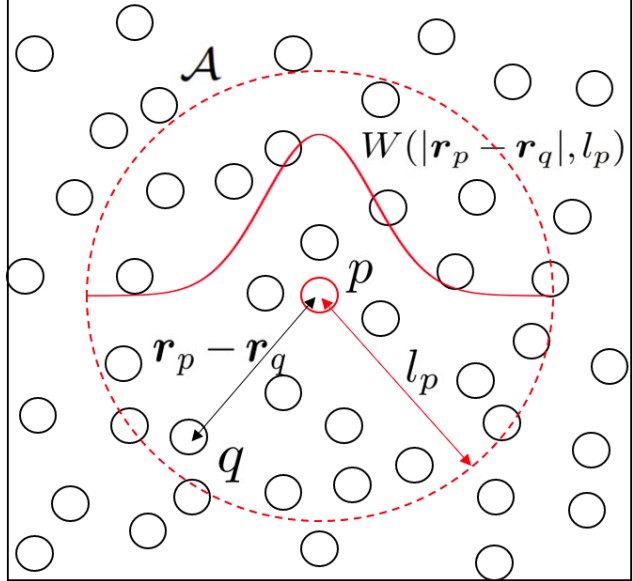

**Figure 1.** Graphical representation of the SPH kernel $W(|\boldsymbol{r}_p - \boldsymbol{r}_q|, l_p)$ (solid red line), the smoothing length $l_p$ (red arrow), the particle $p$, the neighbouring particles $q$, the support domain $\mathcal{A}$ (dashed red line) and the distance between any neighbour particle $q$ and the particle $p$ within the support domain $\boldsymbol{r}_p - \boldsymbol{r}_q$ (black arrow). Note that particles are points in space and that their size in this schematic is arbitrary.

tensor ($\boldsymbol{T}$) and the gradient of a vector field ($\boldsymbol{V}$) (Monaghan, 2005) as (see Appendix A for complete derivation) :

$$(\nabla \cdot \boldsymbol{V})_p = \frac{1}{\rho_p} \sum_{q=1}^{N} m_q (\boldsymbol{V}(\boldsymbol{r}_q) - \boldsymbol{V}(\boldsymbol{r}_p)) \cdot \nabla_p W_{pq}, \tag{5}$$

$$(\nabla \cdot \boldsymbol{T})_p = \rho_p \sum_{q=1}^{N} m_q \left( \frac{\boldsymbol{T}(\boldsymbol{r}_q)}{\rho_q^2} + \frac{\boldsymbol{T}(\boldsymbol{r}_p)}{\rho_p^2} \right) \cdot \nabla_p W_{pq}, \tag{6}$$

$$(\nabla \boldsymbol{V})_p = \sum_{q=1}^{N} \frac{m_q}{\rho_q} (\boldsymbol{V}(\boldsymbol{r}_q) - \boldsymbol{V}(\boldsymbol{r}_p)) \otimes \nabla_p W_{pq}. \tag{7}$$

In Eq. (7), $\otimes$ denotes the outer product. $\nabla_p W_{pq}$ is the gradient of the kernel at the coordinate $\boldsymbol{r}_p - \boldsymbol{r}_q$ in the reference frame of

particle $p$ and is written as :

$$\nabla_p W_{pq} = \frac{\boldsymbol{r}_p - \boldsymbol{r}_q}{|\boldsymbol{r}_p - \boldsymbol{r}_q|} \frac{\partial W(|\boldsymbol{r}_p - \boldsymbol{r}_q|, l_p)}{\partial |\boldsymbol{r}_p - \boldsymbol{r}_q|}. \tag{8}$$

Note that $W_{pq}$ is a scalar function and consequently $\nabla_p W_{pq}$ is a vector, the inner product in Eq. (5) is a scalar, the inner product in Eq. (6) is a 2D vector and the outer product in Eq. (7) is a 2D tensor of rank 2. In addition to Eq. (2 - 3), the smoothing





kernel must have the following set of properties to avoid non-physical behaviour and costly computation (Liu and Liu, 2003):

Compact support :               $W(|\boldsymbol{r}_p - \boldsymbol{r}_q|, l_p) = 0$, for $|\boldsymbol{r}_p - \boldsymbol{r}_q| > l_p,$                      (9)

Positive definite :                   $W(|\boldsymbol{r}_p - \boldsymbol{r}_q|, l_p) \geq 0,$                              (10)

Monotonically decreasing : $\dfrac{\partial W(|\boldsymbol{r}_p - \boldsymbol{r}_q|, l_p)}{\partial(|\boldsymbol{r}_p - \boldsymbol{r}_q|)} \leq 0,$                         (11)

Symmetric :                       $W(|\boldsymbol{r}_p - \boldsymbol{r}_q|, l_p) = W(-|\boldsymbol{r}_p - \boldsymbol{r}_q|, l_p),$             (12)

Differentiable :               $\dfrac{\partial^n W(|\boldsymbol{r}_p - \boldsymbol{r}_q|, l_p)}{\partial(|\boldsymbol{r}_p - \boldsymbol{r}_q|)^n} \exists,$                       (13)

where $\exists$ stands for *exist*. In the above, differentiable means that the kernel derivatives exist up to the highest order present in the equations. Finally, to ensure the consistency of the SPH method approximations to the $n^{th}$ order, all kernel moments of order 1 to $n$ need to vanish. In practice, the consistency conditions are satisfied when the number of neighbouring particles is sufficiently large to be evenly distributed in the domain of influence (Fraga Filho, 2019). Note that, at the boundaries, the domain of influence of the particle is truncated making it impossible to satisfy the kernel moments equations. This phenomenon

is referred as the particle inconsistency and leads to poorer approximations of physical properties. No clear solutions to this problem are proposed in the literature yet.

## 2.2   Momentum and continuity equations

Following Plante et al. (2020), we consider sea-ice to behave as a two-dimensional granular material described by the 2D momentum equation (neglecting the Coriolis and sea surface tilt terms):

$\rho_i h \dfrac{D\boldsymbol{u}}{Dt} = \nabla \cdot \boldsymbol{\sigma} + \boldsymbol{\tau},$                                                                 (14)

where $\rho_i$ is the ice density, $h$ is the mean ice thickness (ice volume over an area), $\boldsymbol{u} = u\hat{\boldsymbol{x}} + v\hat{\boldsymbol{y}}$ is the ice velocity vector, $\boldsymbol{\sigma}$ is the vertically integrated internal stress tensor acting in the $\hat{\boldsymbol{y}}$ direction on a face with a unit outward normal pointing in the $\hat{\boldsymbol{x}}$ direction, $\boldsymbol{\tau}$ is the sum of water drag and surface air stress and $\frac{D}{Dt} = \frac{\partial}{\partial t} + \boldsymbol{u} \cdot \nabla$ is the Lagrangian derivative operator. We neglected the Coriolis and sea surface tilt force in the momentum equation to make it easier to validate the model and study the

ice arch formation. Note that using the Lagrangian derivative operator naturally incorporates the advection of momentum in the ice dynamics — a term that is typically neglected for most continuum based Eulerian sea-ice models. The surface air stress and the water stress can be written using bulk formulation as (McPhee, 1979):

$\boldsymbol{\tau} = \rho_a C_a |\boldsymbol{u}_a - \boldsymbol{u}|(\boldsymbol{u}_a - \boldsymbol{u}) + \rho_w C_w |\boldsymbol{u}_w - \boldsymbol{u}|(\boldsymbol{u}_w - \boldsymbol{u}),$                          (15)

$\approx \rho_a C_a |\boldsymbol{u}_a|(\boldsymbol{u}_a) + \rho_w C_w |\boldsymbol{u}_w - \boldsymbol{u}|(\boldsymbol{u}_w - \boldsymbol{u}),$                          (16)





where $\rho_a$ and $\rho_w$ are air and water densities, $\boldsymbol{u}_a$ and $\boldsymbol{u}_w$ are air and water velocity vectors, $C_a$ and $C_w$ are the air and water drag coefficients and where $\boldsymbol{u}$ is neglected in the formulation of the wind stress since $\boldsymbol{u} \ll \boldsymbol{u}_a$. The continuity equations for the mean ice thickness $h$ and the ice concentration $A$ can be written as:

$$\frac{\mathrm{D}h}{\mathrm{D}t} + h\nabla \cdot \boldsymbol{u} = 0, \tag{17}$$

$$\frac{\mathrm{D}A}{\mathrm{D}t} + A\nabla \cdot \boldsymbol{u} = 0, \tag{18}$$

where the thermodynamic source terms are omitted.

### 2.3 Constitutive laws

The constitutive relations for the viscous-plastic ice model with an elliptical yield curve, a normal flow rule and tensile strength can be written as (Beatty and Holland, 2010):

$$\sigma_{ij} = 2\eta\dot{\epsilon}_{ij} + \left[(\zeta - \eta)\dot{\epsilon}_{kk} - \frac{P_r(1 - k_t)}{2}\right]\delta_{ij}, \tag{19}$$

$$\dot{\epsilon}_{ij} = \frac{1}{2}\left(\frac{\partial u_j}{\partial x_i} + \frac{\partial u_i}{\partial x_j}\right) = \frac{1}{2}\left(\nabla\boldsymbol{u} + \nabla\boldsymbol{u}^\mathsf{T}\right), \tag{20}$$

where $\dot{\epsilon}_{ij}$ is the symmetric part of the strain-rate tensor, $\zeta$ and $\eta$ are the non-linear bulk and shear viscosities, $P_r$ is the replacement pressure, $k_t$ is the tensile strength factor and $\delta_{ij}$ is the Kronecker delta. Following Bouchat and Tremblay (2017) we write :

$$\zeta = \frac{P(1 + k_t)}{2\Delta^*}, \tag{21}$$

$$\eta = \frac{\zeta}{e^2} = \zeta\left(\frac{2S}{P(1 + k_t)}\right)^2, \tag{22}$$

$$\Delta^* = \max(\Delta, \Delta_{min}), \tag{23}$$

$$\Delta = \left[(\dot{\epsilon}_{11}^2 + \dot{\epsilon}_{22}^2)(1 + e^{-2}) + 4e^{-2}\dot{\epsilon}_{12}^2 + 2\dot{\epsilon}_{11}\dot{\epsilon}_{22}(1 - e^{-2})\right]^{1/2}, \tag{24}$$

where $P = P^*h\exp(-C(1 - A))$ is the ice strength (Hibler, 1979), $P^*$ and $C$ are respectively the ice compressive strength and ice concentration parameters, $S$ is the ice shear strength and $e$ is the ellipse aspect ratio. In the limit where the strain rates $\dot{\epsilon}$ go to zero, $\zeta$ and $\eta$ would tend to infinity. To avoid this situation, the deformation $\Delta$ is capped to $\Delta_{min} = 2 \times 10^{-9}\mathrm{s}^{-1}$. Using the $\Delta^*$ formulation, the replacement pressure $P_r$ can be written as

$$P_r = P\frac{\Delta}{\Delta^*}, \tag{25}$$

which ensures that the stresses are zero when the strain rates are zero.



## 2.4 Governing differential equations: SPH framework

To solve ice dynamic system of equations in the SPH framework, equations involving spatial derivatives (Eqs. 14 - 17 - 18 - 20) are reformulated using Eqs. (5 - 6 - 7) with the particle subscripts $p$ and $q$ (see Fig. 1) and a temporal evolution for the ice particle position is defined:

$$\frac{\mathrm{D}\boldsymbol{x}_p}{\mathrm{D}t} = \boldsymbol{u}_p, \qquad\qquad \text{Momentum} \qquad (26)$$

$$\rho_i h_p \frac{\mathrm{D}\boldsymbol{u}_p}{\mathrm{D}t} = \rho_p \sum_{q=1}^{N} m_q \left( \frac{\boldsymbol{\sigma}_q}{\rho_q^2} + \frac{\boldsymbol{\sigma}_p}{\rho_p^2} \right) \cdot \nabla_p W_{pq} + \boldsymbol{\tau}_p, \qquad\qquad \text{Momentum} \qquad (27)$$

$$\frac{\mathrm{D}h_p}{\mathrm{D}t} + \frac{h_p}{\rho_p} \sum_{q=1}^{N} m_q (\boldsymbol{u}_q - \boldsymbol{u}_p) \cdot \nabla_p W_{pq} = 0, \qquad\qquad \text{Continuity} \qquad (28)$$

$$\frac{\mathrm{D}A_p}{\mathrm{D}t} + \frac{A_p}{\rho_p} \sum_{q=1}^{N} m_q (\boldsymbol{u}_q - \boldsymbol{u}_p) \cdot \nabla_p W_{pq} = 0, \qquad\qquad \text{Continuity} \qquad (29)$$

$$(\dot{\epsilon}_{ij})_p = \frac{1}{2} \left[ \left( \sum_{q=1}^{N} \frac{m_q}{\rho_q} (\boldsymbol{u}_q - \boldsymbol{u}_p) \otimes \nabla_p W_{pq} \right) + \left( \sum_{q=1}^{N} \frac{m_q}{\rho_q} (\boldsymbol{u}_q - \boldsymbol{u}_p) \otimes \nabla_p W_{pq} \right)^{\mathsf{T}} \right]. \qquad \text{Constitutive} \qquad (30)$$

It is important to make the distinction between the intrinsic ice density $\rho_i$ and the particle densities $\rho_p$. For consistency reasons with the standard VP rheology, we consider the following definition of density independent of ice concentration in contrast with previous work (Wang et al., 1998; Ji et al., 2005; Staroszczyk, 2017) (see results section for discussion):

$$\rho_p = \rho_i h_p. \qquad\qquad (31)$$

By formulating density as Eq. (31), the continuity Eq. (28) has the same form as the more commonly used continuity density equation (Monaghan, 2012) :

$$\frac{\mathrm{D}\rho_p}{\mathrm{D}t} = -\rho_p \nabla \cdot \boldsymbol{u}_p = \sum_{q=1}^{N} m_q (\boldsymbol{u}_p - \boldsymbol{u}_q) \cdot \nabla_p W_{pq}, \qquad\qquad (32)$$

except for the fact that the divergence of the velocity field is scaled by the ice material density $\rho_i$ ($\frac{\mathrm{D}\rho_p}{\mathrm{D}t} = \rho_i \frac{\mathrm{D}h_p}{\mathrm{D}t}$). Note that since the particle density $\rho_p$ is independent of the concentration, the particle concentration $A_p$ is a quantity that measures the compactness of the floes at the particle location, but does not relate to the amount of ice carried by a particle. With this formulation, the concentration can be interpreted as the probability of ice floes carried by a particle to come in contact with ice floes of another particle (and repel each other) within the unresolved area $\Delta A_p$.





---

**Algorithm 1** Sea-ice SPH

---

**Require:** Domain shape and boundaries, Spatial resolution, Total integration time
  initialize particle and boundary according to input
  **for** $i = 0$ **to** $IntegrationTime$ **do**
    $nInteraction \leftarrow nearestNeighbourParticleSearch$
    **for** $j = 0$ **to** $nInteraction$ **do**
      $kernel \leftarrow smoothingFunctionCalculation$
      $internalForce \leftarrow kernel$
    **end for**
    **for all** $particles$ **do**
      $externalForce$
      $physicalQuantities \leftarrow (externalForce, internalForce)$
      $density \leftarrow iceThickness$
      $smoothingLength \leftarrow density$
    **end for**
    $timeStep \leftarrow smoothingLength$
    monitor particle interaction statistics
    output
  **end for**

---

180 ## 2.5 Numerical approach

Following Hosseini et al. (2019), we use a second order predictor-corrector scheme to evolve in time the SPH ice system of equation (see algorithm 1 below). This integration scheme takes a given function $f$ (here $f$ can be $\boldsymbol{x}$, $\boldsymbol{u}$, $A$ and $h$) and used a predictor step to calculate its value $f^{n+1/2}$ at time $t = (n + \frac{1}{2})\Delta t$ (where $\Delta t$ is the time step) followed by a correction step to calculate the solution $f^{n+1}$ at time $t = (n+1)\Delta t$ from $f^{n+1/2}$:

185
$$f_p^{n+1/2} = f_p^n + \frac{\Delta t}{2}\frac{\mathrm{D}f_p^n}{\mathrm{D}t} + O(\Delta t^2), \tag{33}$$

$$f_{p\,\mathrm{corrected}}^{n+1/2} = f_p^n + \frac{\Delta t}{2}\frac{\mathrm{D}f_p^{n+1/2}}{\mathrm{D}t}, \tag{34}$$

$$f_p^{n+1} = 2f_{p\,\mathrm{corrected}}^{n+1/2} - f_p^n + O(\Delta t^3). \tag{35}$$

Following Lemieux and Tremblay (2009), a simple 1D model taking into account only the viscous term — the most restrictive condition — leads to the following stability criterion:

190
$$\Delta t \leq \frac{\rho_i h l_{min}^2}{\eta_{max}} = \frac{e^2 \rho_i l_{min}^2 \Delta_{min}}{P^*(1 + k_t)}, \tag{36}$$

where $l_{min}$ is the minimum smoothing length across all the particles.



## 2.6 Particle interactions

Following Rhoades (1992), we use the bucket search algorithm parallelized using shared memory multiprocessing (OpenMP) to find all the neighbours of each particle in favour of the explored tree algorithm (Cavelan et al., 2019) which involve pointers and complex memory structure that are not easy to manipulate in OpenMP.

After the neighbour search, the interactions between pairs of particles are computed using the Wendland $C^6$ kernel — Wendland kernels have the best stability properties for wavelengths smaller than the smoothing kernel (Dehnen and Aly, 2012) — which is written as:

$$W(|\boldsymbol{r}_p - \boldsymbol{r}_q|, l_p) \quad = W_{C^6}(R) \quad = \alpha_d \begin{cases} (1-R)^8(32R^3 + 25R^2 + 8R + 1), & 0 \le R < 1, \\ 0, & R \ge 1, \end{cases} \tag{37}$$

$$\frac{\partial W(|\boldsymbol{r}_p - \boldsymbol{r}_q|, l_p)}{\partial |\boldsymbol{r}_q - \boldsymbol{r}_p|} = \frac{\partial W_{C^6}(R)}{\partial |\boldsymbol{r}_q - \boldsymbol{r}_p|} = \alpha_d \begin{cases} -22R(16R^2 + 7R + 1)(1-R)^7 \frac{\kappa}{l_p}, & 0 \le R < 1, \\ 0, & R \ge 1, \end{cases} \tag{38}$$

where $\alpha_d$ is a normalization factor depending on the dimension of the problem. Note that $R\,(= \kappa|\boldsymbol{r}_p - \boldsymbol{r}_q|/l_p)$ is the normalized distance between particles in the referential $\boldsymbol{r}_p - \boldsymbol{r}_q$. Consequently, we always integrate from 0 to $l_p$ (the smoothing length) independently of the kernel instead of 0 to $\kappa l_p$ as shown in (Liu and Liu, 2010). The constant $\alpha_d$ becomes $\frac{78\kappa^2}{7\pi l^2}$ in 2D, with a factor of $\kappa^2$ different from the usual definition. Note that the scaling factor $\kappa$ has a value of 1 for the Wendland $C^6$ kernel. The choice of kernel was validated using stability tests with six different kernels including the original Gaussian kernel (Gingold and Monaghan, 1977), a quartic spline Gaussian approximation (Liu and Liu, 2010), a quintic spline Gaussian approximation (Morris et al., 1997), a quadratic kernel (Johnson and Beissel, 1996) and the Wendland $C^2$, $C^4$ and $C^6$ kernels (Wendland, 1995).

## 2.7 Smoothing length

The smoothing or correlation length is a key element of SPH and has a direct influence on the accuracy of the solution and the efficiency of the computation. For instance, if $l_p$ is too small, there may not be enough particles in the support domain violating the kernel moments requirements. If the smoothing length $l_p$ is too large, all the local properties of particles would be smoothed out over large number of neighbours and the computation time would increase with the number interactions. In two dimensions the optimal number of neighbours interacting with any particle $p$ should be about 20 to balance the precision and the computational cost (Liu and Liu, 2003). We therefore implement a variable smoothing length that evolves in time and space to maintain this approximate number of neighbours. To this end, we keep the mass of particles constant in time and evaluate the smoothing length from the particle density. Note that keeping the mass of a particle constant has the advantage of ensuring mass conservation. This assumption is justified in our case since we are only interested in sea-ice dynamics and ridging change the area cover by ice floes but not their mass. However, fixing the ice mass is only valid when neglecting the thermodynamics and need to be modified for synoptic scale simulation.





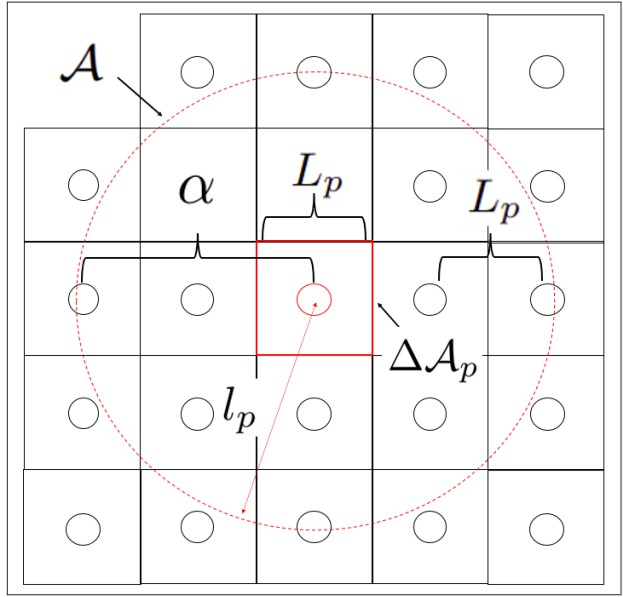

**Figure 2.** Graphical representation of the initial position of the particles and the relevant parameter for the smoothing length evolution : the ice area carried by the particle $\Delta\mathcal{A}_p$ (solid red square), the parameter $\alpha$ ($= 2$ in this schematic for visibility), the support domain $\mathcal{A}$ (dashed red line), the smoothing length $l_p$ (red arrow) and the initial distance between particle $L_p$. Black circles are neighbouring particle $q$ and the red circle is the current particle $p$. Note that, as for the figure 1, the particle sizes in this schematic are also arbitrary.

The initial mass of a particle is defined from the ice area it represents within its support domain ($\Delta\mathcal{A}_p$ in Fig. 2). To avoid creating porosity in the medium, we divide the space in equal square area ($= L_p^2$) that covers the whole domain. Since we want approximately 20 neighbours for every particle, we introduce $\alpha$ ($= 3$ in all simulations) a parameter that stands for the approximate number of particles desired in any direction within the support domain. The parameter $\alpha$ can also be interpreted

as the proportionality constant between the particle spacing $L_p$ and the smoothing length $l_p$. Note that to increase accuracy of the particle approximation, $\alpha$ can be increased by any desired factor (see Fig. 2). The mass carried by a particle is therefore written as :

$$m_p = \Delta\mathcal{A}_p \rho_i h_{0p} = L_p^2 \rho_i h_{0p}, \tag{39}$$

where $h_{0p}$ is the initial mean thickness of the particle. The smoothing length is then updated at each time step diagnostically

from:

$$l_p = \alpha L_p = \alpha \sqrt{\frac{m_p}{\rho_p}}. \tag{40}$$





**Table 1.** Physical parameters used in ridging and arch simulations.

| Parameter | Symbol | Value | Unit |
|---|---|---|---|
| Ice concentration parameter | $C$ | 20 | - |
| Ice compressive strength | $P^*$ | 27.5 | $\mathrm{kN} \cdot \mathrm{m}^{-2}$ |
| Air density | $\rho_a$ | 1.3 | $\mathrm{kg} \cdot \mathrm{m}^{-3}$ |
| Water density | $\rho_w$ | 1026 | $\mathrm{kg} \cdot \mathrm{m}^{-3}$ |
| Ice density | $\rho_i$ | 900 | $\mathrm{kg} \cdot \mathrm{m}^{-3}$ |
| Wind drag coefficient | $C_a$ | $1.2 \times 10^{-3}$ | - |
| Water drag coefficient | $C_w$ | $5.5 \times 10^{-3}$ | - |
| Minimal total deformation | $\Delta_{min}$ | $2 \times 10^{-9}$ | $\mathrm{s}^{-1}$ |

Values of the parameter used for the simulations are the same as the one presented in (Williams et al., 2017) to facilitate comparison in the results section.

The smoothing length $l_p$ is capped to 10 times its initial value when the particle density tends to zero. This capping prevents conservation of mass for density lower than 1% of its initial value (see Eq. (39)). We justify this capping because such small densities do not affect the ice dynamics.

## 2.8 Boundary treatment

We implemented the boundary treatment of Monaghan and Kajtar (2009) because of its simplicity, versatility and low computational cost. The boundaries are set up by placing stationary particles with fixed smoothing length $l_b$ and a mass $m_b$ equal to the average ice particle mass $m_p$. The boundary smoothing length $l_b$ is chosen in a way that only one layer of ice particles initially interact with the boundary (this makes $l_b$ resolution dependent). The boundary particles are (equally) spaced apart by

a factor one quarter of their smoothing length ($l_b/4$). In this manner, all ice particles $p$ within a support domain $l_b$ will interact with approximately four boundary particles (denoted by the subscript $b$) at a time resulting in a net normal repulsive force $\boldsymbol{F_{Np}}$:

$$\boldsymbol{F_{Np}} = \sum_{b=1}^{N_b} \kappa_n \frac{(\boldsymbol{r}_p - \boldsymbol{r}_b)}{|\boldsymbol{r}_p - \boldsymbol{r}_b|^2} W_{pb} \frac{2m_b}{m_p + m_b}, \tag{41}$$

that is added to their momentum equation. In Eq. (41), $\kappa_n$ is a constant with units of $[\mathrm{kg} \cdot \mathrm{m}^4 \cdot \mathrm{s}^{-2}]$ used to adjust the repulsion

strength and is also simulation dependent because it needs to counterbalance the particle acceleration, and prevent them from escaping the domain. This free parameter is not suited for complex pan-arctic simulations, but is sufficient in our idealize experiment study. A free-slip boundary condition in all simulations, i.e., no tangential friction force between boundary particle and ice particle is applied.





## 3 Results

### 3.1 Plastic wave propagation


We first compare the plastic wave speed for the VP dynamic equations with and without the SPH approximations. To this end, we do a perturbation analysis for a one-dimensional case with a fixed sea-ice concentration ($A = 1$). In this case, the 1D SPH sea-ice dynamic equations (Eqs. 26 - 29) form a system of three equations and three unknowns ($x$, $u$ and $h$) :

$$\frac{\mathrm{D}x_p}{\mathrm{D}t} = u_p \tag{42}$$


$$\frac{\mathrm{D}u_p}{\mathrm{D}t} = \Gamma \sum_{q=1}^{N} \frac{m_q}{\rho_i^2} \left( \frac{1}{h_q} + \frac{1}{h_p} \right) \frac{x_{pq}}{|x_{pq}|} \frac{\partial W}{\partial x_{pq}} + \tau_p, \tag{43}$$

$$\frac{\mathrm{D}h_p}{\mathrm{D}t} = -\frac{1}{\rho_i} \sum_{q=1}^{N} m_q (u_q - u_p) \frac{x_{pq}}{|x_{pq}|} \frac{\partial W}{\partial x_{pq}}, \tag{44}$$

where $x_{pq}$ is a short form for $x_p - x_q$ and $\Gamma = \frac{P^*}{2} \left[ \pm (e^{-2} + 1)^{1/2} - 1 \right]$. In the above, we made use of the following 1D normal stress for convergent plastic motion (see Gray, 1999; Williams et al., 2017, for 1D normal stress derivation):

$$\sigma = \sigma_{xx} = \frac{P^*}{2} \left[ \pm (e^{-2} + 1)^{1/2} - 1 \right] h = \Gamma h. \tag{45}$$

Linearizing around a mean state ($\bar{u} = 0$ and $\bar{h} = h_0$), considering small perturbations ($\delta x$, $\delta u$ and $\delta h$) and ignoring $2^{nd}$ order term, we obtain:

$$\frac{\mathrm{D}\delta x_p}{\mathrm{D}t} = \delta u_p \tag{46}$$

$$\frac{\mathrm{D}\delta u_p}{\mathrm{D}t} = \frac{\Gamma}{\rho_i} \sum_{q=1}^{N} \Delta \mathcal{A}_q \frac{\bar{x}_{pq}}{|\bar{x}_{pq}|} \left( \frac{-1}{h_0} (\delta h_q + \delta h_p) \frac{\partial W}{\partial \bar{x}_{pq}} + 2(\delta x_p - \delta x_q) \frac{\partial^2 W}{\partial \bar{x}_{pq}^2} \right), \tag{47}$$

$$\frac{\mathrm{D}\delta h_p}{\mathrm{D}t} = -h_0 \sum_{q=1}^{N} \Delta \mathcal{A}_q \frac{\bar{x}_{pq}}{|\bar{x}_{pq}|} (\delta u_q - \delta u_p) \frac{\partial W}{\partial \bar{x}_{pq}}, \tag{48}$$

where $\Delta \mathcal{A}_q = \frac{m_q}{\rho_i h_0} = \frac{m_q}{\rho_q}$ (Eq. 4) and where we have used the binomial expansion $\frac{1}{h} = \frac{1}{h_0 + \delta h} \approx \frac{1}{h_0}(1 - \frac{\delta h}{h_0})$. Assuming perturbations have a wavelike solution of the form $\delta f = \hat{f} \exp(i(k\bar{x} - \omega t))$ — where $i$ is the imaginary number, $k$ is the wavenumber





and $\omega$ is the angular velocity — the set of equations in the reference frame following the ice motion reduces to:

$$\hat{x} = \frac{i}{\omega}\hat{u}, \tag{49}$$

$$\hat{u} = \frac{i\Gamma}{\omega\rho_i}\sum_{q=1}^{N}\mathcal{A}_q\frac{\bar{x}_{pq}}{|\bar{x}_{pq}|}\left(\left[-\frac{\hat{h}}{h_0}(1+\exp(-ik\bar{x}_{pq}))\right]\frac{\partial W}{\partial \bar{x}_{pq}} + 2\hat{x}(1-\exp(-ik\bar{x}_{pq}))\frac{\partial^2 W_{pq}}{\partial \bar{x}_{pq}^2}\right), \tag{50}$$

$$\hat{h} = -\frac{ih_0\hat{u}}{\omega}\sum_{q=1}^{N}\mathcal{A}_q\frac{\bar{x}_{pq}}{|\bar{x}_{pq}|}(\exp(-ik\bar{x}_{pq})-1). \tag{51}$$

Note that since the ice is initially at rest, the Lagrangian and the Eulerian frameworks are equivalent. For large enough wavelength (so that the perturbation can be resolved across multiple particles with high accuracy i.e., $\lambda \geq l_p$ and $N \to \infty$), the summations can be written as integrals, i.e., $\sum_{q=1}^{N}\mathcal{A}_q\frac{\bar{x}_{pq}}{|\bar{x}_{pq}|}$ becomes $\int_{-\infty}^{\infty}\mathrm{d}\bar{x}_{pq}$. Taking advantage of the kernel properties — i.e., all moments higher than 0 vanish — we can write Eqs. 50 - 51 as:

$$\hat{u} = \frac{-i\Gamma}{\omega\rho_i}\int_{-\infty}^{\infty}\left(\frac{\hat{h}}{h_0}\frac{\partial W}{\partial \bar{x}_{pq}}+2\hat{x}\frac{\partial^2 W_{pq}}{\partial \bar{x}_{pq}^2}\right)\exp(-ik\bar{x}_{pq})\mathrm{d}\bar{x}_{pq} = \frac{\Gamma}{\omega\rho_i}\left(\frac{\hat{h}}{h_0}k+i2k^2\hat{x}\right)\tilde{W}, \tag{52}$$

$$\hat{h} = -\frac{ih_0\hat{u}}{\omega}\int_{-\infty}^{\infty}\exp(-ik\bar{x}_{pq})\frac{\partial W}{\partial \bar{x}_{pq}}\mathrm{d}\bar{x}_{pq} = \frac{h_0\hat{u}k}{\omega}\tilde{W}, \tag{53}$$

where the integrals have been converted to Fourier transform using $\mathcal{F}(\frac{\partial W}{\partial \bar{x}_{pq}}) = ik\mathcal{F}(W) = ik\tilde{W}$. Finally, combining Eqs. (49 - 52 - 53), the phase speed for the plastic wave ($\frac{\omega}{k}$) can be written as:

$$c_{\mathrm{SPH}} = \frac{\omega}{k} = \pm\tilde{W}\sqrt{-\frac{\Gamma}{\rho_i}\left(\frac{2}{\tilde{W}}-1\right)}. \tag{54}$$

For wavelengths much larger than the smoothing length ($\lambda \propto \frac{1}{k} \gg l_p$), the Fourier transform of the kernel tends to 1 ($\tilde{W} \approx 1$) and the SPH formulation reduces to the Viscous-Plastic theory without SPH approximations (see for instance Williams et al., 2017), i.e.:

$$c_{\mathrm{VP}} = \pm\sqrt{-\frac{\Gamma}{\rho_i}}, \tag{55}$$

with a plastic wave propagation speed $c_{\mathrm{VP}} \approx 5.7$ [m·s$^{-1}$] for typical sea-ice parameters (see Table 1). Consequently, a major difference of SPH with the FDM framework is that the plastic wave speed is dispersive with a phase velocity $c_{\mathrm{SPH}}$ that is dependent on the wavelength and the smoothing length. In general, only the plastic waves with a wavelength between approximately 1 and 11 times the smoothing length will have their travelling speed modified by more than 1%. More specifically, in the limit where the wavelength $\lambda$ approaches the smoothing length $l_p$, the plastic wave speed increases in the SPH framework





for a maximum value of $\approx 6.7 \, [\mathrm{m \cdot s^{-1}}]$ (see Fig. 3 panel a). Note that for wavelength smaller than the smoothing length, the
summations in Eqs. (52- 53) cannot be written as integrals but the particles still respond partially to the perturbations. This
sometimes leads to the tensile and the zero-energy modes instabilities (Swegle et al., 1995). As mentioned above, Dehnen and
Aly (2012) showed that Wendland kernels, can diminish the tensile instability and the pairing of particles. A deeper analysis
of unresolved waves ($\lambda < l_p$) in the context of sea-ice SPH dynamic equations is beyond the scope of the current study.

For the more general case when the base state allows for a variable concentration (linearized around a mean state $\bar{A} = A_0$)
and considering the classical — denoted by a superscript $C$ — particle density definition ($\rho_p^{\mathrm{C}} = \rho_i h_p A_p$) used by Wang et al.
(1998); Ji et al. (2005); Staroszczyk (2017), the plastic wave speed becomes:

$$c_{\mathrm{A,SPH}}^{\mathrm{C}} = \pm\tilde{W}\sqrt{-\frac{\Gamma^*}{\rho_i}\left(CA_0 - 3 + \frac{2}{\tilde{W}}\right)}, \tag{56}$$

where $\Gamma^* = \Gamma \exp(-C(1 - A_0))$. We argue that the plastic wave speed $c_{\mathrm{A,SPH}}^{\mathrm{C}}$ obtained with the classical density definition
does not converge (see Fig. 3 panel b) to the Viscous-Plastic theory, $c_{\mathrm{A,VP}}$, derived from FDM (see Williams et al., 2017, for
derivation):

$$c_{\mathrm{A,VP}} = \pm\sqrt{-\frac{\Gamma^*}{\rho_i}\left(CA_0 + 1\right)}, \tag{57}$$

because the ice concentration is taken into account in both the definition of $\rho_p^{\mathrm{C}}$ and implicitly in the definition of the average
thickness $h_p$. When we consider the new formulation of particle density independent of concentration as proposed above (Eq.
31) the wave speed equation becomes:

$$c_{\mathrm{A,SPH}} = \pm\tilde{W}\sqrt{-\frac{\Gamma^*}{\rho_i}\left(CA_0 - 1 + \frac{2}{\tilde{W}}\right)}, \tag{58}$$

which reduces to the FDM VP theory (Eq. 57) when the wavelength is large compared to the smoothing length (see Fig. 3 panel
c). However, the classical density definition is not wrong, Wang et al. (1998); Ji et al. (2005); Staroszczyk (2017) used different
formulation of the continuity equation in their model which makes our perturbation analysis only valid in the current study. In
a similar manner as for the plastic wave speed with a fixed concentration (Eq. 54), the wave speed $c_{\mathrm{A,SPH}}$ (Eq. 58) is dispersive
and the wavelength between 1 and 11 times the smoothing length are those that are mostly affected (more than 1%). However,
in this case, the plastic wave speed is damped for wavelengths close to the smoothing length for mean concentration state
higher than 0.1. Note that while the plastic wave speed is defined for all A, it does not have a physical meaning for $A < 0.85$
since there are negligible ice-ice interactions.

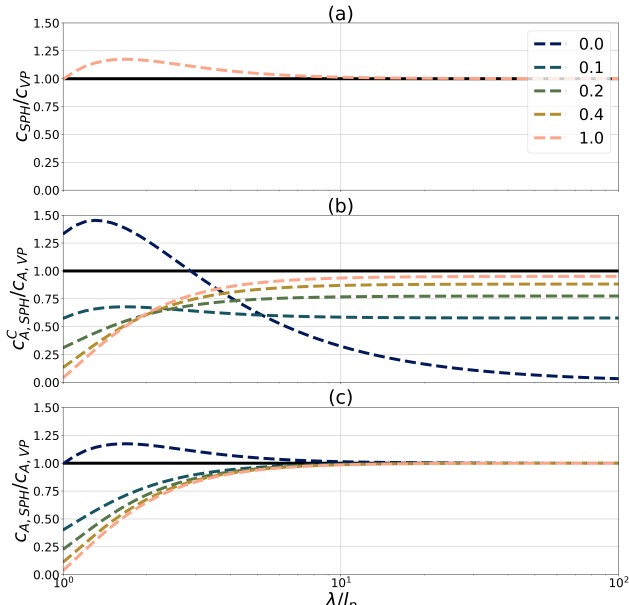

**Figure 3.** SPH plastic wave speed as a function of the normalized wavelength ($\lambda/l_p$) for the Wendland $C^6$ kernel. Panel a) show the classical VP rheology with fixed concentration (Eq. 54) normalized by the FDM plastic wave speed with fixed concentration (Eq. 55), panel b) show the classical VP rheology with a variable concentration and the density definition $\rho_p^C = \rho_i h_p A_p$ (Eq. 56) normalized by the FDM plastic wave speed with a variable concentration (Eq. 57) and panel c) show the classical VP rheology with a variable concentration and the density definition $\rho_p = \rho_i h_p$ (Eq. 58) normalized by the FDM plastic wave speed with a variable concentration (Eq. 57). Different homogeneous base state of concentration $A_0$ are shown varying from 0 to 1.

## 3.2 Ridging experiments

We validate our implementation of the SPH model (with the new definition of particle density $\rho_p$) in a 1D ridging experiment for which we have an analytical solution (see Williams and Tremblay, 2018, for derivation):

$$-\frac{d\sigma}{dx} = \rho_a C_a |u_a| u_a \implies \frac{dh}{dx} = \frac{2\rho_a C_a |u_a| u_a}{P^*(\sqrt{e^{-2}+1}+1)}, \tag{59}$$

i.e., a linear profile in thickness with a slope proportional to the square of the wind velocity and inversely proportional to the ice strength. We consider a rectangular domain of 1000 by 2000 [km] including the boundary (the ice field is 1900 [km] to
ensure that no particle escape on the open side) with 37240 particles, an initial homogeneous smoothing length $l_p$ of 21.429 [km] (spacing $l_p/\alpha = 7.14$ [km]) and a smaller — to limit boundary effect — boundary particle smoothing length $l_b$ of 4 [km] (spacing $l_b/4 = 1.0$ [km]) to represent the wall (see Fig. 4). Particles are initialized with an average thickness $h = 1$ [m] and a concentration $A = 1$. They are forced against the wall by a constant unidirectional wind of 5 [m·s$^{-1}$]. Note that the water drag force is removed in the simulation for a faster convergence to the steady state which enables higher resolution — a water current
of 0 [m·s$^{-1}$] would slow down the ice and the ridge formation since it is driven by the advection speed. The Coriolis force should normally also have to be considered with this domain size and classical polar latitude — the Rossby number is $\mathcal{O}(10^{-2})$





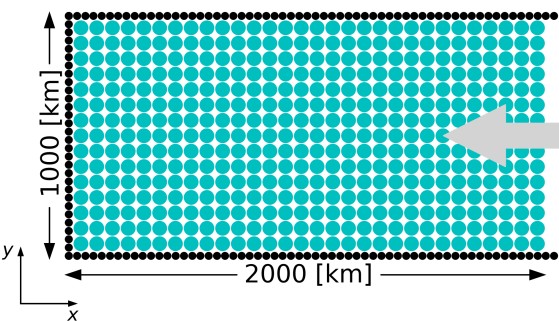

**Figure 4.** Idealized domain of the ridging experiment. The blue circles represent the ice particles and the black ones are the boundary particles. The grey arrow shows the wind forcing.

—, but is neglected in this idealized experiment to conserve the symmetry of the solution and compare it to the theoretical 1D equation ( Eq. 59). In results presented below (Fig. 5 - 6), the particles properties are averaged over a grid of approximately 10 by 5 [km] cells for plotting purposes. Results show that the simulated thickness field converges to the analytical solution

(within an error of $\approx 1\%$) after $\approx 5$ days with a slope of $1.33 \times 10^{-3}$ [m·km$^{-1}$], compared with $1.34 \times 10^{-3}$ [m·km$^{-1}$] for the theory. Artifacts are observed close to the boundary where the repulsive force prevent the particle from reaching the "wall". Additionally, when a particle comes into contact with the boundary with a certain inertia (due to the $1/\boldsymbol{r}$ dependence of the boundary force), we observe oscillations in the motion of particles which can propagate far in the domain ( e.g., Fig. 5 panel a, at $x \approx [50, 300]$ [km] and $t = [30, 45]$ [h]). The oscillations are damped and the energy is dissipated by the rheology term with

time until an equilibrium is reached. A more physical boundary treatment is beyond the scope of this study.

    We also repeated the ridge experiment with the same forcing and total sea-ice volume but letting the sea ice concentration evolve with time, specifically with an initial average thickness and concentration of $h = 0.5$ [m] and $A = 0.5$, to ensure that both $h$ and $A$ covary in time such that $\frac{h}{A}$ remains constant in the MIZ until significant ice interactions take place (see Fig. 5 panel b). To accomplish this, the domain was extended to 4000 [km] (3800 [km], excluding the boundaries) and the initial

particles spacing changed from 7.14 [km] to 10.0 [km] for a corresponding initial smoothing length $l_p$ of 30.0 [km] and total number of particles of 38000. In this configuration, the model converges to a steady state solution in $\approx 10$ days with a slope $1.36 \times 10^{-3}$ [m·km$^{-1}$], in agreement with theory within an error of $\approx 1\%$ (see Fig. 5 panel b). Results at $x = 300$ [km], away from boundary effects, show that (as desired) thickness and concentration evolve coherently — $h/A$ is constant in time — before ice concentration reaches $\approx 85\%$ (see Fig. 6 panel a). At that point ($t \approx 22$ [h]), ice-ice interactions emerge and the

ridging process starts ($\frac{\mathrm{d}(h/A)}{\mathrm{d}t} > 0$). One key difference with the simulation initialized at $A = 1$ is a thickness build-up (above 1 m) at the edge of the ridge in the marginal ice zone (MIZ). At this location, the continuity equation for sea ice concentration is capped while that of the mean ice thickness remains continuous. This results in a local increase in ice thickness to $\approx 1.1$ [m]. This process is akin to the wave radiation drag in the MIZ (Sutherland and Dumont, 2018). A detailed analysis of simulations in simple convergent ice flow in the MIZ with ice concentration close to 100 % will be considered in future work.

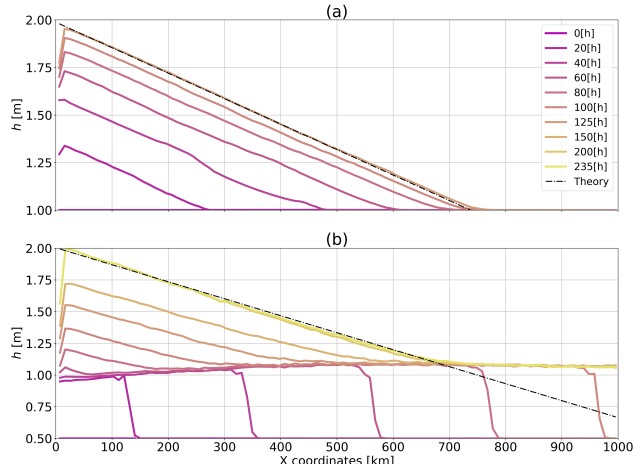

**Figure 5.** Temporal evolution of simulated sea-ice thickness along the central horizontal line of the domain for (a) the ridge experiment initialized with a concentration $A = 1$ and average thickness $h = 1$ and (b) the ridge experiment initialized with a concentration $A = 0.5$ and average thickness $h = 0.5$. The wall is located at $x = 0$ and the wind speed is $-5\hat{\boldsymbol{x}}$ [m · s$^{-1}$]. The theory follows Eq. (59).

In the ridge building phase, the speed of advance of the ridge front increases until a maximum concentration is reached after $\approx 70$ [h] (see Fig. 6 panel c). Subsequently, the ice drift speed reduces and the rate of advance of the ridge slows down. When the ice thickness gradient is in balance with the surface wind stress (after $\approx 200$ [h]), $\frac{\mathrm{d}(h/A)}{\mathrm{d}t}$ reaches steady state. Overall, we can observe three stages in the ridge formation (see Fig. 6). First, a rapid compaction stage, when ice particles are drifting close to their free drift speed since the ice strength is weak. Second, a transition stage between $A \approx 0.85 - 1.00$ when ridging occurs

in the MIZ analogous to the wave radiation drag mentioned above. Third, a ridging stage with changes in ice thickness that are about one order of magnitude higher than during the transition stage. Note that the amplitude of oscillations between particles within the domain or at the boundaries in ridging experiments diminishes when incorporating the water drag (a damping term). The water drag also lead to a longer time scale to reach steady state, since the ice drift speed is slower.

### 3.3   Arch experiments

We next compare the SPH approach with the FDM and DEM sea-ice model in a second well-studied idealized experiment: the ice arches formation. To this end, we run the SPH model in an idealized domain representing the Nares Strait (see Fig. 7) with an upstream reservoir 5 times the length of the channel ($L$) to minimize boundary confinement effect without sacrificing the spatial resolution.

     The set of simulations uses a domain with $L = 60$ [km]. The initial condition for ice thickness, concentration and velocity

are $h = 1$ [m], $A = 1$ and $\boldsymbol{u} = 0$ [m · s$^{-1}$]. The ice is forced with a constant unidirectional wind of $-7.5$ [m · s$^{-1}$] in the $\hat{\boldsymbol{y}}$-direction and ocean current is fixed to $\boldsymbol{u}_w = 0$ [m · s$^{-1}$]. The corresponding surface stress is $\approx 0.04$ [kN · m$^{-2}$] and the total integrated stress at the entry of the channel is slightly smaller than $P^*$ ($\int_0^{5L} \tau_a \mathrm{d}x = 26.325$ [kN · m$^{-1}$]). We use a weaker wind than what is common in Nares Strait ice arches simulations ($\approx 10$ [m · s$^{-1}$]) to limit the ridging phase prior to the formation


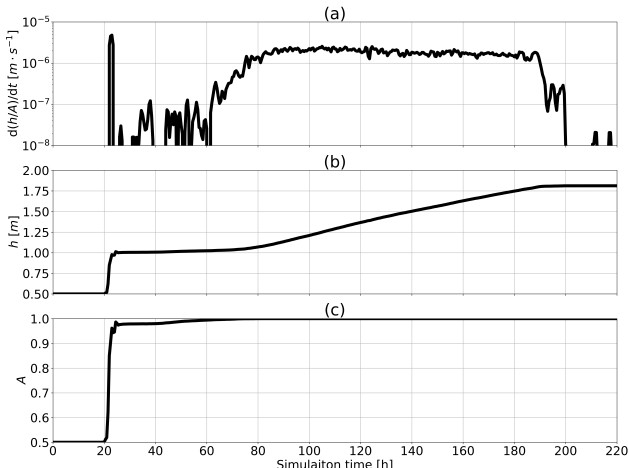

**Figure 6.** Evolution in time of a) the thickness normalized by concentration rate of change in time $\frac{\mathrm{d}(h/A)}{\mathrm{d}t}$, b) the average thickness $h$ and c) the concentration $A$ at $x = 300$ [km]. The rate of change in time is computed from $\frac{\mathrm{d}f}{\mathrm{d}t}(x,t) = \frac{f(x,t+\Delta t) - f(x,t-\Delta t)}{2\Delta t}$.

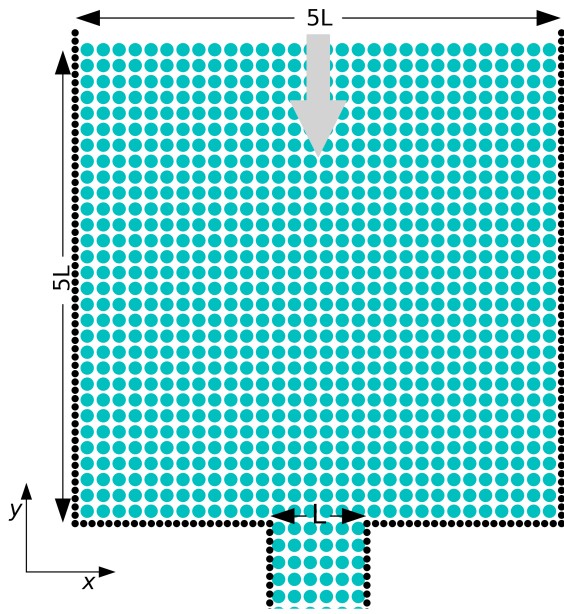

**Figure 7.** Idealized domain of the ice arch experiments. The blue circles represent the ice particles and the black ones are the boundary particles. The grey arrow shows the wind forcing.

of the ice arch. In this experiment, we limit ourselves to ice with no tensile strength ($k_t = 0$) and a shear strength of 6.875

[kN · m$^{-2}$], i.e., an ellipse aspect ratio of 2.





We first test whether the SPH approach has the same sensitivity to the relative size of particle with respect to the channel width as in DEM (Damsgaard et al., 2018). Results showed that no stable arch can be formed with the specified forcing for all particle diameter size tested (7.5, 5, 3.75 [km]) (see ice velocity field Fig.9). Instead, a "continuous" slow flow of ice is present in the channel. The discontinuity at the entry of the channel visible in the concentration, thickness and velocity fields (Fig. 9)

can be interpreted as intermittent (unstable) ice arch formation. Also, we noted that larger particles are not more prone to ice jam than smaller ones. This is contrary to what is know from granular material theory and to results from Damsgaard et al. (2018) that show a transition from stable to no ice arch formation for floe sizes ranging from approximately one quarter to one sixteenth of the strait width. We explain this difference between SPH and DEM from the continuum description of the ice dynamics equation which describes the ice strength as a function of ice concentration and mean thickness, not on the particle

size. Even though the increase in resolution — or particle size — has no effect on the arch stability, it enables smaller fractures resolution that are visible at the entrance of the channel (see $\epsilon_I$ and $\epsilon_{II}$ Fig. 8). In our SPH model, the stress invariants $\sigma_I$ and $\sigma_{II}$ shows oscillation patterns in regions where ice is in the viscous regime (see the tree-like structure in the normal and shear stress fields in Fig. 8). We hypothesized that those are associated with over-damped viscous waves occurring with small movement of the particle undergoing viscous deformation. Those structures are not symmetric, despite symmetrical initial

conditions, because of the domino effect between interacting viscous waves. Note that they are absent from the strain-rate fields since viscous deformation are extremely small. They are also absent in sea-ice model based on a continuum approach (Dumont et al., 2009; Dansereau et al., 2017; Plante et al., 2020), but these tree-like structures are qualitatively similar to the stress structure between floes observed in DEM (e.g., Damsgaard et al., 2018, Fig. 5c).

Second, we explored the ability of the model to produce stable ice arches. To this end, we reduce the total integrated surface

stress at the entry of the channel to 13.146 $[\mathrm{kN} \cdot \mathrm{m}^{-1}]$ (or wind speed of 5.3 $[\mathrm{m} \cdot \mathrm{s}^{-1}]$) to a value below the ice compressive strength ($P^*$) to avoid completely ridging north of the channel and jump immediately in the arch-forming stage. In this case, results show a clear stable arch (see Fig. 10) with a shape that is qualitatively similar to the one presented by Dansereau et al. (2017); Plante et al. (2020); West et al. (2022). The formation of a stable arch in an SPH model is possible with the standard shear strength ($e = 2$), contrary to continuum model that required an increase in shear strength (Dumont et al., 2009;

Dansereau et al., 2017; Plante et al., 2020, e.g.,). This suggests that SPH has a different sensitivity of ice arching to the ellipse aspect ratio $e$ and ice thickness $h$. With a no slip boundary condition and the same default yield curve (same $P^*$ and ellipse aspect ratio $e$), preliminary results suggest that no arches form — the pack is undeformed — and instead a higher surface wind stress is required to form an arch. Overall, this shows that SPH is able to capture large-scale features coming from small-scale interactions. The simulation of a stable ice arch (Fig. 10) also shows how SPH can fracture and create discontinuities in the

ice field as seen in DEM models. This behaviour is similar to that of the elastic-decohesive sea-ice constitutive of Schreyer et al. (2006) or the FEM model of Rampal et al. (2016). Finally, in the SPH framework, a lead or polynya can be defined by an absence of particles for leads larger than particle size — akin to DEM — or by particles with reduced concentration for sub-particle size leads — akin to FDM.

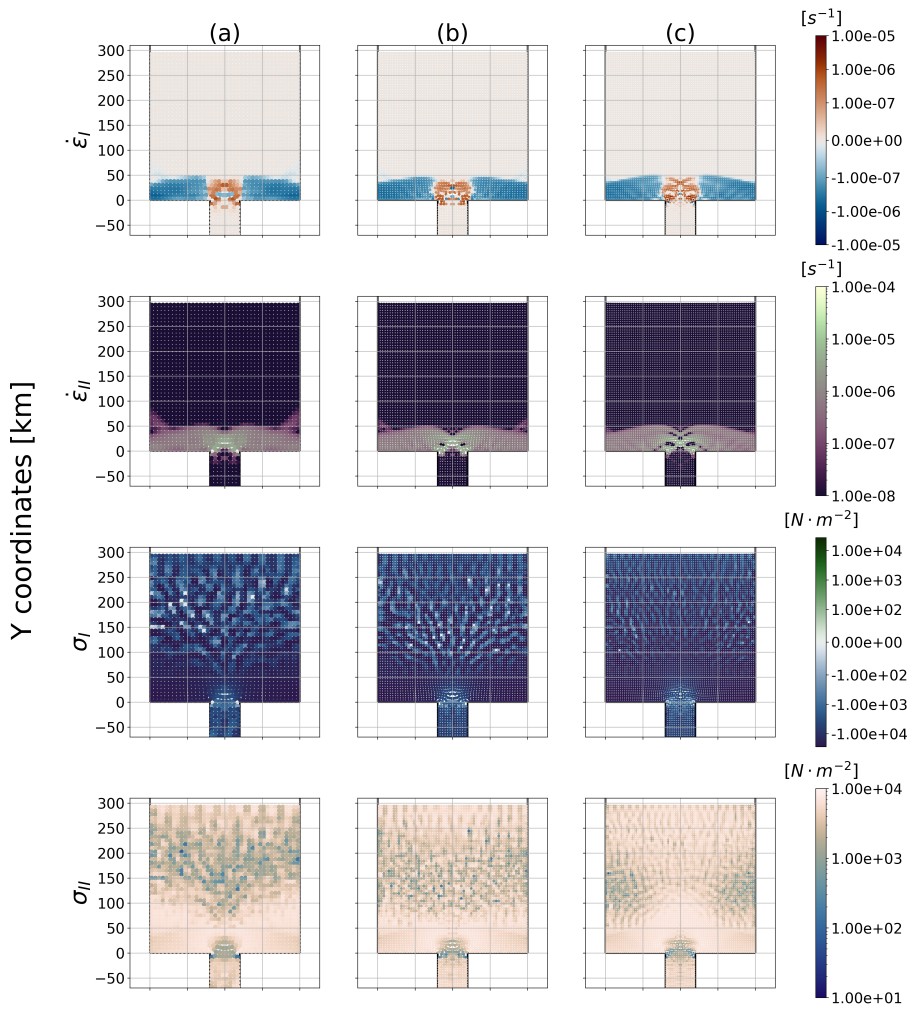

**Figure 8.** Strain rate and stress invariants ($\dot{\epsilon}_I$, $\dot{\epsilon}_{II}$, $\sigma_I$, $\sigma_{II}$) at time $t = 24$ [h] for an initial particle spacing of a) 7.5, b) 5 and c) 3.75 [km] (8, 12 and 16 particles can fit in the strait respectively) and the initial total integrated surface stress at the entry of the channel is 26.325 [kN · m$^{-1}$].

## 4 Discussion and Conclusion

In this paper, we have presented a first implementation of the Viscous-Plastic rheology with an elliptical yield curve and normal flow rule in the framework of SPH with the long-term goal of simulating synoptic scale sea-ice dynamics. We have described the basics of the SPH approach and how the sea-ice dynamic equations can be formulated in this framework along with the implementation of key components of the numerical method such as the smoothing length, the kernel, the boundaries and the time integration technique. We proposed a different definition of the particle density and showed that the more commonly





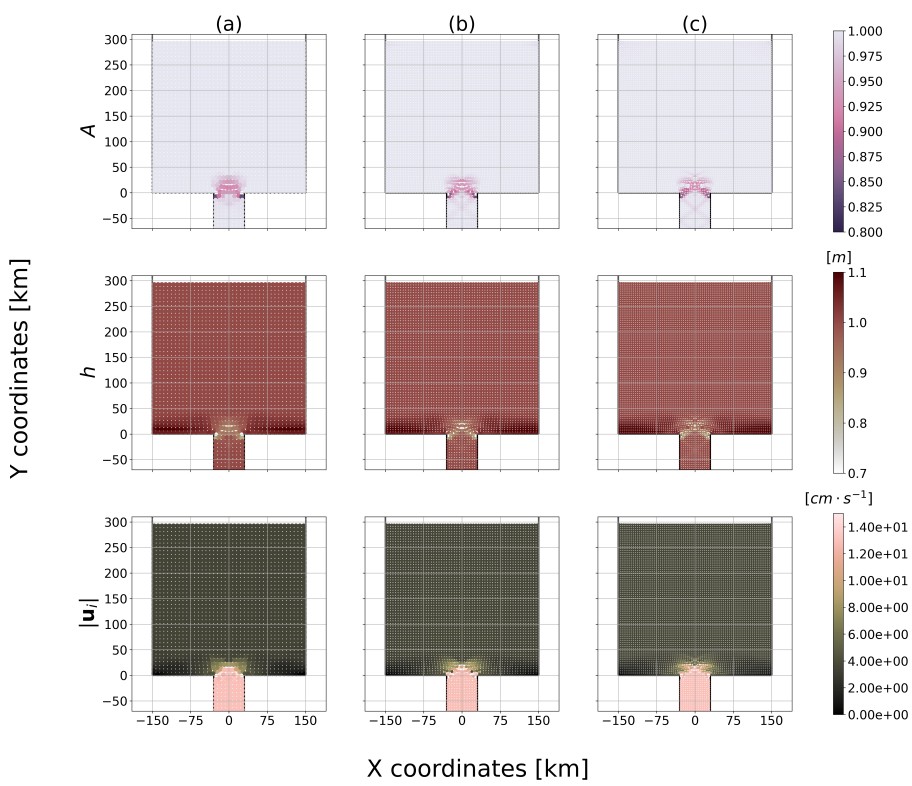

**Figure 9.** Ice concentration, thickness and total velocity ($h$, $A$, $|\boldsymbol{u}_i|$) at time $t = 24$ [h] for an initial particle spacing of a) 7.5, b) 5 and c) 3.75 [km] (8, 12 and 16 particles can fit in the strait respectively) and the initial total integrated surface stress at the entry of the channel is 26.325 [kN · m$^{-1}$].

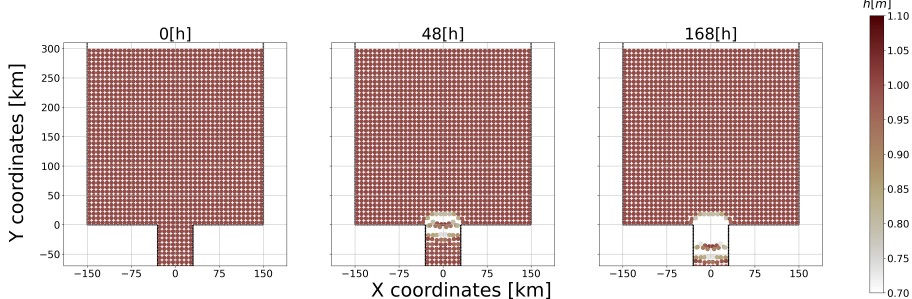

**Figure 10.** Thickness field at time $t = 0, 48, 168$ [h] for an initial particle spacing of 7.5 [km] and a total integrated stress at the entry of the channel of 13.146 [kN · m$^{-1}$].

used density definition involving the ice concentration when used together with the average ice thickness leads to erroneous plastic wave speed propagation. A particle density definition independent of the ice concentration corrects this and leads to results that are inline with the VP theory. The SPH model thus developed is in excellent agreement (error of $\approx 1\%$) with an analytical solution of the VP ice dynamic for a simple 1D ridging experiment. The approximations used at the core of the SPH





framework, result in a dispersive plastic wave speed in the medium — contrary to its FDM counterpart — which is dependent
on the smoothing length (or resolution) and the choice of the kernel. The plastic wave speed is mostly affected for wavelengths
11 times the smoothing length and lower.

From the simple ridging experiment with fixed sea-ice concentration ($A = 1$), we observe nonphysical damped oscillations
that propagate in the domain associated with our choice of boundary conditions. The conclusions drawn from our simulations
are robust to the choice of boundary conditions. Nevertheless, this behaviour needs to be removed for a proper simulation of
sea-ice near coastlines. The ridging experiment with an initial ice concentration below 100% showed that continuity equations
for concentration and thickness evolve coherently until a concentration of 85%. At that point, SPH particles start to ridge locally
in the MIZ in addition to the wall where the maximum stress is located. This effect is not observed in continuum approach and
is presumably related to particle collisions in converging motion.

When compared to other numerical framework, the SPH model is able to reproduce stable ice arches in an idealized domain
of a strait with an ellipse aspect ratio of 2 and a wind forcing of 5.3 $[\mathrm{m \cdot s^{-1}}]$, contrary to other continuum approaches that
require higher material shear strength. However, when using a stronger wind field of 7.5 $[\mathrm{m \cdot s^{-1}}]$, no stable arches are formed
when increasing the particles size in the strait (stable arches are only achieved when increasing particle average thickness).
We concluded that the number of particles in the strait does not influence the formation of ice arches contrary to DEM and
is analogous to an increase in resolution in a continuum framework : a larger number of particles influence the number of
fractures that can form and the resolution of fine-scale structures. The stress fields produced by the SPH model in the channel
experiment show tree-like pattern upstream of the channel where there are low total deformations. This is not observed in FDM
experiment but is qualitatively similar to tensile stress network exhibited in DEM (Damsgaard et al., 2018) that comes from
individual contact force between the ice floes and is hypothesized to be associated with damped viscous sound waves.

Even though we successfully implemented the standard sea-ice Viscous-Plastic rheology with an elliptical yield curve and
a normal flow rule in an SPH framework, the current model does not outperform classical FDM model. In fact, there are
inherent difficulties and instabilities in SPH that do not exist in FDM. It is known that the SPH framework trade consistency
— i.e., the ability to correctly represent a differential equation in the limit of an infinite number of points with a null spacing
between them — for stability , which gives the SPH a distinct feature of working well for many complicated problems with
good efficiency, but less accuracy. However, the classical formulation of SPH used and described in the present work does
not usually respect zeroth-order consistency because of the unstructured particle position in space(see Belytschko et al., 1998,
section 3 for derivation). Nevertheless, consistency can be improved at the expense of computational cost (Chen and Beraun,
2000; Liu et al., 2003) by reformulating the SPH core approximation (Eq. 1). Also, boundary description has been identified as
a weak point of the SPH framework since prescribing a Dirichlet or Neumann boundary condition is not as straightforward as
in continuum approaches and preventing particle penetration through a boundary is still a challenging task (Liu and Liu, 2010)
and the SPH consistency is usually at its worst at the boundary because the support domain is truncated. In the present study,
a proper physical representation of the boundary was not adopted and the boundary treatment was chosen for its numerical
simplicity and should be modified in future work. Other major issues with SPH are the zero-energy modes and the tensile
instability previously mentioned. The zero-energy modes can be found in FDM and FEM and they correspond to modes at





which the strain energy calculated is erroneously zero (Swegle et al., 1995). The tensile instability results in particle clumping or
nonphysical fractures in the material. In the present work, we adopted a different kernel from the usual Gaussian spline to avoid
those instabilities, but other methods such as the independent stress point (Dyka and Ingel, 1995; Chalk et al., 2020), artificial
short length repulsive force (Monaghan, 2000), particle repositioning (Sun et al., 2018), adaptive kernel (Lahiri et al., 2020),
etc. can be used if more stabilization is needed. For example, at smaller scale, SPH simulation of ice in uniaxial compression
was improved by a simplified finite difference interpolation scheme (Zhang et al., 2017). More specifically for sea-ice model,
Kreyscher et al. (2000) pressure closure is not well suited for long simulation. Indeed, particle can still move when they are
in the viscous state but, have low internal ice pressure because of the replacement pressure scheme. Consequently, particles
could pass through each other resulting in erroneous location of the parameters carried. Finally, using SPH for sea-ice modules
in grid-based continuum global climate model (GCM) complicates the coupling with ocean and atmosphere components since
particle quantities need to be converted on a grid and vice versa.

Nevertheless, SPH also has interesting properties that could be exploited. For example, SPH can be used with little change
for problems involving several fluids whether liquid, gas, or dust fluids (Monaghan, 2012). This feature could be exploited
in the creation of a general approach for all components of a GCM (atmosphere, ocean and sea-ice). The method developed
is also a proper option for nowcasting sea-ice prediction because only the ice dynamics need to be considered in nowcasting
applications and the model has a good ability to carry the ice property in space. SPH can fracture and transitions from the
continuum to fragments seamlessly, which is the main reason for our investigation of the method for sea-ice dynamics. The
elastic behaviour assumed for sea-ice in certain rheology can be associated to the weak compressibility inherent in the classical
formulation of SPH. Finally, the SPH discretization of the continuum into particles enables the implementation of several new
features. For example, angular momentum to individual floes (or pack of floes) can be added to take into account rotation along
LKFs. A direct measure of the concentration from the number of particles within a support domain (this takes advantages
of already computed number of neighbours and help ensuring the desired number of neighbours in converging flow) can be
computed. A subscale parametrization of floe-floe contact force (this short length repulsive force could also help for the tensile
instability) can be implemented. A varying floe size distribution can be incorporated by varying the mass carried by a particle
for a given particle density.

For future work, a more physical treatment of the boundary conditions should be investigated — e.g., using the immerse
boundary method (Tu et al., 2018) with a fixed grid for the boundary and an interpolation scheme to apply force on the particle
to simulate the grounding of sea-ice near the coast. In order to use the model for pan-Arctic simulations, the Coriolis and sea
surface tilt force along with the treatment of the thermodynamics source and sink terms should be implemented in the SPH
framework (see preliminary work by Staroszczyk, 2018). In addition, the parallelization of the code should be improved in
order to bring the computational time down to a value comparable to that of an FDM model. Finally, while there still is a
significant amount of work to be completed before SPH can be used in large-scale climate simulations, the method shows
promises and deserves further investigations and development.





*Code and data availability.* Output data from the SPH sea-ice model simulations along with a version of the model used and the analyzing programs are available at https://doi.org/10.5281/zenodo.6950156

**Appendix A: Vector operators in SPH**

Vector operators take different forms in the SPH framework because they only operate on the smoothing kernel $W$ and they need to ensure symmetric interactions between particles. The following subsections show the demonstrations to obtain the relevant one to our study.

**A1  Divergence of a vector**

First, the divergence of vector needs to be changed into a form that can be symmetrized. To do so, we use the identity of the divergence of a scalar function times a vector and chose the scalar function to be the density as follow:

$$\nabla \cdot \boldsymbol{V} = \frac{1}{\rho} \left( \nabla \cdot (\rho \boldsymbol{V}) - \boldsymbol{V} \cdot \nabla \rho \right). \tag{A1}$$

Now applying the integral interpolant approximation (1) to the divergence term ($\nabla \cdot (\rho \boldsymbol{V})$) and to the density ($\rho$) gives:

$$\nabla \cdot (\rho \boldsymbol{V}) = \int_{\mathcal{V}} \nabla' \cdot (\rho' \boldsymbol{V'}) W \, d\boldsymbol{r'} = \int_{\mathcal{V}} \nabla' \cdot (\rho' \boldsymbol{V'} W) d\boldsymbol{r'} - \int_{\mathcal{V}} \rho' \boldsymbol{V'} \cdot \nabla' W d\boldsymbol{r'}, \tag{A2}$$

$$\rho = \int_{\mathcal{V}} \rho' W d\boldsymbol{r'}. \tag{A3}$$

In the above equations, the prime quantities represents the surrounding values. Note that the kernel is the only function that depends on both primed and non-primed position as defined at (1). Using divergence theorem the first term in (A2) can be cancelled :

$$\int_{\mathcal{V}} \nabla' \cdot (\rho' \boldsymbol{V'} W) d\boldsymbol{r'} = \int_{\mathcal{S}} (\rho' \boldsymbol{V'} W) \cdot d\boldsymbol{s'} = 0, \tag{A4}$$





since the integration surface $\mathcal{S}$ encompassing the volume $\mathcal{V}$ is arbitrary and the kernel $W$ has the compact support property

(Eq. 9). Applying the particle approximation (4) to the Eqs. (A2) - (A3), we obtain:

$$(\nabla \cdot (\rho \boldsymbol{V}))_p = -\sum_q m_q \boldsymbol{V}_q \cdot \nabla_q W_{pq} = \sum_q m_q \boldsymbol{V}_q \cdot \nabla_p W_{pq}, \tag{A5}$$

$$\rho_p = \sum_q m_q W_{pq}, \tag{A6}$$

where we used the identity $\nabla_p = -\nabla_q$ and $p$ and $q$ represent the current particle and neighbour. Finally, substituting the last

two Eqs. (A5 - A6) in (A1) gives the desired form of the operator:

$$(\nabla \cdot \boldsymbol{V})_p = \frac{1}{\rho_p} \left( \sum_q m_q \boldsymbol{V}_q \cdot \nabla_p W_{pq} - \boldsymbol{V}_p \cdot \nabla_p \sum_q m_q W_{pq} \right) \tag{A7}$$

$$= \frac{1}{\rho_p} \left( \sum_q m_q (\boldsymbol{V}_q - \boldsymbol{V}_p) \cdot \nabla W_{pq} \right). \tag{A8}$$

### A2    Divergence of a 2D tensor field

Note that in the following demonstration, the Einstein summation convention is used to simplify the calculation and the tensor

representation. We start with the divergence of a 2D tensor divided by the density:

$$\frac{\partial}{\partial x_i} \left( \frac{T_{ij}}{\rho} \right) = \frac{1}{\rho} \frac{\partial T_{ij}}{\partial x_i} - \frac{T_{ij}}{\rho^2} \frac{\partial \rho}{\partial x_i}. \tag{A9}$$

Reorganizing the terms gives :

$$\frac{\partial T_{ij}}{\partial x_i} = \rho \left[ \frac{\partial}{\partial x_i} \left( \frac{T_{ij}}{\rho} \right) + \frac{T_{ij}}{\rho^2} \frac{\partial \rho}{\partial x_i} \right]. \tag{A10}$$

Now applying the interpolant approximation (1) to the first term in the bracket leads to :

$$\frac{\partial}{\partial x_i} \left( \frac{T_{ij}}{\rho} \right) = \int_{\mathcal{V}} \frac{\partial}{\partial x_i'} \left( \frac{T_{ij}'}{\rho'} \right) W \, d\boldsymbol{r}' \tag{A11}$$

$$= \int_{\mathcal{V}} \frac{\partial}{\partial x_i'} \left( \frac{T_{ij}'}{\rho'} W \right) d\boldsymbol{r}' - \int_{\mathcal{V}} \left( \frac{T_{ij}'}{\rho'} \right) \frac{\partial W}{\partial x_i'} d\boldsymbol{r}'. \tag{A12}$$

As for the divergence of a vector demonstration (section A1), the first integral above vanish by using the divergence theorem

and applying the particle approximation gives:

$$\left( \frac{\partial}{\partial x_i} \left( \frac{T_{ij}}{\rho} \right) \right)_p = -\sum_q \left( m_q \frac{(T_{ij})_q}{\rho_q^2} \right) \frac{\partial W_{pq}}{\partial (x_i)_q} = \sum_q \left( m_q \frac{(T_{ij})_q}{\rho_q^2} \right) \frac{\partial W_{pq}}{\partial (x_i)_p}. \tag{A13}$$





Substituting this in the Eq. (A10) and using the equality A6 we get the following expression :

$$\left(\frac{\partial T_{ij}}{\partial x_i}\right)_p = \rho_p \left[\sum_q \left(m_q \frac{(T_{ij})_q}{\rho_q^2}\right)\frac{\partial W_{pq}}{\partial (x_i)_p} + \frac{(T_{ij})_p}{\rho_p^2}\frac{\partial}{\partial (x_i)_p}\left(\sum_q m_q W_{pq}\right)\right] \tag{A14}$$

$$= \rho_p \left[\sum_q m_q \left(\frac{(T_{ij})_q}{\rho_q^2} + \frac{(T_{ij})_p}{\rho_p^2}\right)\frac{\partial W_{pq}}{\partial (x_i)_p}\right] \tag{A15}$$

$$= \rho_p \sum_q m_q \left(\frac{\boldsymbol{T}_q}{\rho_q^2} + \frac{\boldsymbol{T}_p}{\rho_p^2}\right)\cdot \nabla_p W_{pq}, \tag{A16}$$

which is the form presented at Eq. (6).

## A3  Gradient of a vector field

To demonstrate the Eq. (7) we first write:

$$\nabla(\boldsymbol{V}1) = 1\nabla\boldsymbol{V} + \boldsymbol{V}\cdot\nabla 1 \tag{A17}$$

$$= \nabla\boldsymbol{V} - \boldsymbol{V}\cdot\nabla 1. \tag{A18}$$

Recall that one property of the kernel is that its zeroth-order moment equals 1:

$$M_0 = \int_{\mathcal{V}} W(\boldsymbol{r} - \boldsymbol{r}', l_p)d\boldsymbol{r}' = 1 \tag{A19}$$

$$= \sum_q \frac{m_q}{\rho_q} W_{pq}. \tag{A20}$$

Now substituting it with the ones in the expression A18 and using the particle approximation (4):

$$(\nabla\boldsymbol{V})_p = \frac{\partial}{\partial (x_i)_p}\sum_q \frac{m_q}{\rho_q}(V_j)_q W_{pq} - (V_j)_p\frac{\partial}{\partial (x_i)_p}\sum_q \frac{m_q}{\rho_q}W_{pq} \tag{A21}$$

$$= \sum_q \frac{m_q}{\rho_q}((V_j)_q - (V_j)_p)\frac{\partial}{\partial (x_i)_p}W_{pq} \tag{A22}$$

$$= \sum_q \frac{m_q}{\rho_q}(\boldsymbol{V}_q - \boldsymbol{V}_p)\otimes\nabla_p W_{pq}, \tag{A23}$$

which is Eq. (7) and where Einstein summation convention was once again used to simplify the derivation.

*Author contributions.* OM coded the model, ran all the simulations, analyzed results and led the writing of the manuscript. BT participated in weekly discussions during the course of the work and edited the manuscript. JFL and MI participated in monthly discussion during the course of the work and edited the manuscript.



*Competing interests.*   The authors declare that they have no conflict of interest.

*Acknowledgements.*   Oreste Marquis is grateful for the support from McGill University, Québec-Océan, and Arctrain Canada. This project was partially funded by a Grant & Contribution from Natural Science and Engineering Research Council Discovery Program and a Grant & Contributions from Environment and Climate Change Canada awarded to Bruno Tremblay.



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
