# Peer review of "Smoothed Particle Hydrodynamics Implementation of the Standard Viscous-Plastic Sea-Ice Model and Validation in Simple Idealized Experiments"

_The Cryosphere, 2022_

## Referee Comment (RC1)

**Review**

**Title: Smoothed Particle Hydrodynamics Implementation of the Standard Viscous-Plastic Sea-Ice Model and Validation in Simple Idealized Experiments**

**Authors: Oreste Marquis, Bruno Tremblay, Jean-François Lemieux Mohammed Islam**

The manuscript presents a realization of the viscous-plastic rheology with an elliptical yield curve and normal flow rule in context of the Smoothed Particle Hydrodynamics meshfree method. The authors describe the basics of the SPH method and derive the formulation of sea ice dynamics within this concept. The SPH method is evaluated on 3 simple test cases. Before reading the manuscript, I did not know the SPH method, but I am very familiar with sea ice dynamics, especially with the viscous-plastic model. I need to say that not knowing the SPH method it was hard to follow the argumentation in the paper (quite technical). Based on the presented manual I could not understand the theoretical derivation of the wave speed in Section 3.1. Looking at the numerical results it is not clear to me if the SPH approach can capture simple sea ice drift. I suggest that the authors simulate standard idealized benchmark tests of the viscous-plastic model to demonstrate that the implementation is correct and simple large-scale drift can be simulated (see main comment). I strongly recommend that the paper is reviewed by another person with a strong background on the SPH method.

Main comment:

I would expect that the SPH realization roughly reflects the drift of the VP model. Therefore, I would first show that simple idealized drift can be reproduced to demonstrate that the implementation is correct and the model does with one expects. In this sense the arch experiment is a bit unfortunate as the SPH method behaves different than the VP model.

I suggest to solve the benchmark problem of Hunke 2001, which has been solved by Danilov et al. 2015 with removed islands. Another candidate for an evaluation would be the benchmark problem of Mehlmann et al. 2021, where the viscous-plastic model has been solved by several institutes. The formation of LKFs has been studied in this paper. It would be of interest to see if your model captures the large-scale drift and produce LKFs, which are large scale features (as the ice arches) that are coming from a small scale.

Further comments:

l. 20 I think the way that the sentence is phrased is not correct. Hunke does not use a classical FDM. A sub-grid discretization is applied for the approximation of the stresses. I would rephrase the sentence to: Traditionally finite difference methods haven been applied to solve the VP model.

l. 188 Why do I need the information on the time step limitation? Can you please add a sentence that explains where this information is used in the ongoing analysis?

l. 257 Please add an equation number to Gamma. The relation is used frequently in the manuscript.

l. 266 Why does it makes sense to assume that the perturbation behaves like a wave solution?

l. 267 'the set of equations' which set?  Please add equation numbers.

l. (49)-(51) Please add a comment where the hat notation is coming from.

l. 273 Please add more details. Why can the summation be written in the integral form. There is no integrant in the integral. Is the integrant 1?

l. 116 Consistency to what?

l. 275 How do you get to the righthand side in equation (52). Can you please add some more steps?

l. 279 It is unclear to me how the wave speed is derived by looking add equations (49)-(53). I stopped reading 3.1 at this line.

l. 351 I think to state that numerical convergence is observed, you need to ensure that even with longer simulations no overshoots occur in fig 5 (a). Is the solution with 200[h] still approached?

l. 338. There are serval definitions of MIZ. How do you define MIZ in your setup?

l 338. How do A and h vary in time? Based on eq. (28) and (29)? Please add some information here.

l.372 Why would you expect a similar sensitivity of the DEM and SPH approach? They are based on different rheologies.

---

## Author Comment (AC4)

**Answers to tc-2022-163 RC1**

September 6, 2023

Note:
- The referee comments are shown in black,
- The authors answers are shown in blue,
- *Quoted texts from the revised manuscript are shown in italic and in dark red.*

\* The exact pages and line numbers in our responses are subjected to change as the revised manuscript is being prepared.
* * *
Review #1

Title: Smoothed Particle Hydrodynamics Implementation of the Standard Viscous-Plastic Sea-Ice Model and Validation in Simple Idealized Experiments

Authors: Oreste Marquis, Bruno Tremblay, Jean-François Lemieux Mohammed Islam The manuscript presents a realization of the viscous-plastic rheology with an elliptical yield curve and normal flow rule in context of the Smoothed Particle Hydrodynamics meshfree method. The authors describe the basics of the SPH method and derive the formulation of sea ice dynamics within this concept. The SPH method is evaluated on 3 simple test cases. Before reading the manuscript, I did not know the SPH method, but I am very familiar with sea ice dynamics, especially with the viscous-plastic model. I need to say that not knowing the SPH method it was hard to follow the argumentation in the paper (quite technical). Based on the presented manual I could not understand the theoretical derivation of the wave speed in Section 3.1. Looking at the numerical results it is not clear to me if the SPH approach can capture simple sea ice drift. I suggest that the authors simulate standard idealized benchmark tests of the viscous-plastic model to demonstrate that the implementation is correct and simple large-scale drift can be simulated (see main comment). I strongly recommend that the paper is reviewed by another person with a strong background on the SPH method.

Main comment:
I would expect that the SPH realization roughly reflects the drift of the VP model. Therefore, I would first show that simple idealized drift can be reproduced to demonstrate that the implementation is correct and the model does with one expects. In this sense the arch experiment is a bit unfortunate as the SPH method behaves different than the VP model. I suggest to solve the benchmark problem of Hunke 2001, which has been solved by Danilov et al. 2015 with removed islands. Another candidate for an evaluation would be the benchmark problem of Mehlmann et al. 2021, where the viscous-plastic model has been solved by several institutes. The formation of LKFs has been studied in this paper. It would be of interest to see if your model captures the large-scale drift and produce LKFs, which are large scale features (as the ice arches) that are coming from a small scale.

The benchmark problems suggested are interesting and would be good tests in future versions of the model. Currently, however, the no slip boundary condition is not implemented in the model;  the physical representation of the boundaries in the  SPH framework  is challenging  and could be a subject of a separate article . In its present form, a normal repulsive force on the ice is applied at the boundary and consequently, only a free slip boundary condition is permitted.

To address this comment, we will instead show the results from the 1D-SIM McGill, a standard viscous-plastic model, in the same ridging experiment and compare with that of the SPH (see Figure 5 of the revised manuscript or below). The SPH and the standard VP model are in agreement. Note that we only show at the steady state because both simulations behave differently in the transient state since we needed to reinsert the water drag to avoid convergence problems with 1D-SIM.

[Figure]

Further comments:

l. 20 I think the way that the sentence is phrased is not correct. Hunke does not use a classical FDM. A sub-grid discretization is applied for the approximation of the stresses. I would rephrase the sentence to: Traditionally finite difference methods haven been applied to solve the VP model.

This sentence is revised as suggested by the reviewer. The new sentence at l.21 now reads : "
*The more commonly used constitutive laws are the standard Viscous-Plastic model (Hibler, 1979) or modifications thereof (e.g., Elastic-Viscous-Plastic or EVP and Elastic-Plastic-*

*Anisotropic or EPA; Hunke and Dukowicz, 1997; Tsamados et al., 2013). They are typically discretized on an Eulerian mesh using finite-difference method (FDM)."*

l. 116 Consistency to what?

We refer to the consistency of a discretization of a PDE. This is clarified in the revised manuscript.

*The new sentence at l.539 reads: " Finally, to ensure the consistency of the discretization of PDEs (as defined in Belytschko 1998) of the SPH method…"*

l. 188 Why do I need the information on the time step limitation? Can you please add a sentence that explains where this information is used in the ongoing analysis?

The time step limitation is used to set the time step for the SPH model. It can be calculated from a characteristic ice velocity and the radius of influence of the kernel in the model. This was clarified in the revised manuscript.

*The new sentence at l.167 now read: " The stability criterion imposes a strict limitation on the time step ($\sim 10^{-4}$ to $10^{-2}$ seconds for particles of radius between 1 and 10 kilometres) that cannot be avoided by pseudo-time step with a solver because, in the SPH framework, particles are irregularly placed and move around at each time step. This makes the parallelization of particle interactions algorithm mandatory for any practical applications, but the explicit time step avoids possible convergence issues with the use of a numerical solver."*

l. 257 Please add an equation number to Gamma. The relation is used frequently in the manuscript.

We added a reference for the equation for gamma at eq.32 .

l. 266 Why does it make sense to assume that the perturbation behaves like a wave solution?

This is the standard practice when studying the numerical stability of a numerical scheme. One poses a general exponential solution (f_hat exp(i(kx-wt))), solves for k and identifies growing (unstable) mode. This was clarified in the revised manuscript.

*The new sentence at l.251 now reads: " Following Williams et al. (2017), we do a perturbation analysis on the system of equations (34 - 36) and assume a wave solution of the form δf = ˆf exp(i(k x̄ − ωt)) —  where i is the imaginary number, k is the wavenumber, ω is the angular velocity and f is a dummy variable standing for u, x and h — … "*

l. 267 'the set of equations' which set? Please add equation numbers.

Corrected as suggested by the reviewer. The reference at l.254 is now included.

l. (49)-(51) Please add a comment where the hat notation is coming from.

The hat notation is a standard way of expressing the constant coefficient in front of the general exponential (wave) solution.

l. 273 Please add more details. Why can the summation be written in the integral form. There is no integrant in the integral. Is the integrant 1?

The integrant is everything between the integral and the dx since everything not depending on the position has been factored out. This has been clarified in the revised manuscript.

The new sentence at l.259 now reads: *"For large enough wavelengths (so that the perturbation can be resolved across multiple particles with high accuracy i.e., $\lambda \geq lp$ and $N \to \infty$), the summations can be approximated by integrals over the space i.e ..."*

l. 275 How do you get to the righthand side in equation (52). Can you please add some more steps?

As for Equ. 53, the exp() term is distributed and the integrals are the Fourier transforms of the derivative of the kernel (as stated on line 278). Note that at this point we use the W tilde to represent the Fourier transform of the kernel in opposition with W which represents the kernel. This has been clarified by adding steps in the derivation in the revised manuscript on line 265.

l. 279 It is unclear to me how the wave speed is derived by looking add equations (49)-(53). I stopped reading 3.1 at this line.

Equations (49 - 52 - 53) represent 49, 52 and 53 only without 50 and 51. The equation reference has been revised for clarity and, at l.266, we added the sentence: " *Finally, eqs. (37, 40 - 41) represents a system of three equations for three unknowns (ˆx, ˆu, ˆh) that we solve by substitution. This leads to the following form of the phase speed…"*

l. 338. There are several definitions of MIZ. How do you define MIZ in your setup?

We define the MIZ as the area where the sea ice concentration ranges between 0.15 and 0.8.  This was clarified in the revised manuscript at l.334 :  *"… in the marginal ice zone (MIZ), which we define as the area where the sea ice concentration ranges between 0.15 and 0.85 and where low ridging by ice collision occurs... "*

l 338. How do A and h vary in time? Based on eq. (28) and (29)? Please add some information here.

The prognostic variables h and A vary based on the dynamic processes only (i.e. divergence of div (u h)); thermodynamic processes are not considered. We added the following sentence a l. 330 : *This ensures that both h and A covary in time such that h/A remains constant — note that, A and h follow the same continuity equations (15) and (16), or (4) and (5) if omitting the SPH approximations, and therefore should vary identically in time until A reaches 1 — …"*

l. 351 I think to state that numerical convergence is observed, you need to ensure that even with longer simulations no overshoots occur in fig 5 (a). Is the solution with 200[h] still approached?

Lower resolution simulations were run for a much longer time (~200 hr) and were stable. This was clarified in the revised manuscript. The new sentence added at I.320 now reads: *"... after ≈ 5 days — lower resolution simulations (results not shown) were run for a much longer time and also converged to this stable state — with a slope … "*.

 I.372 Why would you expect a similar sensitivity of the DEM and SPH approach? They are based on different rheologies.

The rheology in the DEM emerges from the assumptions made about the shear and normal forces between two colliding boundaries and are usually based on Mohr-Coulomb static friction law. This is different from that of Hibler's standard VP model), as stated by the reviewer. Both DEM and SPH are Lagrangian in nature, and for this reason we expect both to behave in the same manner in some circumstances. Results from Li et al., Herman, 2016; and Damsgaard et al., 2018 in similar idealized domain show similar behavior confirming this.

We have made the sentence softer in the revised manuscript. The new sentence at I.366 now reads*: " We suspect the SPH method to exhibit similar behaviour as DEM methods in certain circumstances even though they have different rheologies, because of their Lagrangian nature. Indeed, the interpretation of the numerical representation of a particle in SPH as a collection of ice floes is also retrieved in DEM (Li et al., 2014) and the two numerical frameworks compute their quantities with one-to-one interactions. Consequently, we first test whether the SPH approach has the same sensitivity to the relative size of particles with respect to the channel width as in DEM (Damsgaard et al., 2018). Results…"*

---

## Author Comment (AC5)

**Answers to tc-2022-163 RC2**

**September 6, 2023**

Note:
- The referee comments are shown in black,
- The authors answers are shown in blue,
- *Quoted texts from the revised manuscript are shown in italic and in dark red.*

\* The exact pages and line numbers in our responses are subjected to change as the revised manuscript is being prepared.
* * *
Review #2

This paper describes the use of SPH for sea ice dynamics. In more detail, it implements VP rheology with elliptical yield curve into a SPH model. This is an interesting exercise and could lead to further work on using SPH on ice dynamics. The paper is worth publishing after some modifications. The comments by this reviewer are, mainly, related to the usefulness and applicability of the method: What is gained by using SPH when compared to FDM or DEM?

On general level:

Paper is very technical and it not easy to follow without a background in SPH. Is there a way to make it easier to read? Considering the readership of TC, effort to do this might increase the number of readers. Even if this reviewer is very familiar with numerical models, cannot go through all the equations of the paper. Authors could consider if such high level of detail needed here or could some parts rely on referencing earlier work? What is new in this description and what is from other sources?

In our opinion, section 2.1 is the only section which is purely theoretical and could be avoided by referencing. It describes rapidly the general concept of SPH and the kernel restrictions and assumptions. However, we feel that omitting it and just referring the theory to previous work for the readers would not give them the tools to understand the reformulation of differential equations of section 2.4 and the importance of the kernel. This is a key component to understand because from it we show that the modifications of the equations for the SPH framework modify the way plastic waves propagate in the medium. Consequently, to address reviewers' comments, we decided to keep section 2.1, but to move it to the Appendix. Sections 2.4 to 2.8 are important for people that want to know how the implementation was done and would like to reproduce or create their own SPH model. If the reviewer insists, more details could be added to specific sections.

The particle size in all simulations is of order of several kilometers. In addition, if the reviewer understands SPH correctly, all quantities in SPH become distributed over even larger area due to smoothing by kernel functions. Discussion on the following five issues in the paper is warranted:

(1) Is your model able to describe discontinuities in the deforming ice field with higher accuracy than typical continuum models (both in the case of opening leads and formation of ridges)?

As stated by the reviewer, the quantities are distributed over a large number of particles according to their smoothing length which also represent the effective resolution. From our dynamic formulation of the smoothing length and the ability of the SPH to move the particles around, we believe that during ridge formation (convergence) the discontinuities will have a high effective resolution because there are a lot of points to capture the ice deformations. On the other hand, in a lead opening (divergence) the edge of the discontinuity can be blurry because of the low number of points, but the shape of the opening is not restricted to a grid which has the advantage to enable smoother edge shape.

This sentence is added at l. 478 in the discussion of the revised manuscript: " … *in space. SPH can fracture and transitions from the continuum to fragments seamlessly since it is not restricted on a grid which also has the advantage of enabling smoother ice edge shape. The ability of the method to move around particles has the interesting property to concentrate them in converging motion increasing the resolution of the model in regions under high stress activity and to scatter them in diverging motion which decreases the resolution in low ice concentration area. This property should result in higher accuracy than typical continuum models. The elastic …*"

(2) Is the resolution of your model higher than typical continuum models?

This depends on the computational power available and the parallelization efficiency of the code. Assuming the same resolution (smoothing lengths equal to the grid cells size) for both techniques the SPH method will adapt its resolution to improve it where the dynamics predict more ice (convergent flow group the particles) and reduce it in areas of low ice concentration (divergent flow scatter the particles). This is different from other continuum approaches where the grid is fixed. Therefore, SPH should improve the overall accuracy because we are usually interested in areas of high ice concentration which have high stress and deformation.

The following sentence is added at l.176 : "*...Note that at its current stage, the model is rudimentary parallelized and a single time step for 40000 particles is of the order of tenths of a second. This implies that the model cannot have a resolution as high as other continuum approaches for the same physical set up. This…*"

(3) is the coarse resolution, or large particle size, due to computational burden?

Yes it is. The SPH method is explicit, which cannot take advantage of the solvers used in FDM like JNFK or Picard-SOR. The SPH explicit formulation forces a really small time step of the order of the hundredths of a second for a smoothing length of 10 km to properly resolve the

plastic-wave propagation. This is around 6 orders of magnitude smaller than the time step used in FDM. The efficiency of SPH comes from its great potential in parallelization which, we believe, could compete with FDM on supercomputers for large simulations.

The following sentence is added in the discussion section 2.4 in the revised at l.167: " *The stability criterion imposes a strict limitation on the time step (~ 10^−4 to 10^−2 seconds for particles of radius between 1 and 10 kilometres) that cannot be avoided by pseudo-time step with a solver because, in the SPH framework, particles are irregularly placed and move around at each time step. This makes the parallelization of particle interactions algorithm mandatory for any practical applications, but the explicit time step avoids possible convergence issues with the use of a numerical solver.*"

(4)    Does it even make sense to decrease the particle size when VP rheology is used?

At the moment, the dynamic formulation of sea ice used in the model is fairly simple and an increase in resolution (or decrease in particle size) is not really useful. The model uses the VP rheology as a test case because it is well known and makes the comparison with previous work easier, but further development using SPH should step away from it.

The following sentence is at l.72 in the revised manuscript:  "*In this work, we use the standard VP sea-ice model with an elliptical yield curve and normal flow rule (Hibler, 1979) as a proof-of-concept. Further development of the  SPH model should consider a broader range of rheologies .  We…*"

(5)    Does an individual particle in your simulation have physical meaning (do they, for example, describe ice floes – you do mention that particle collisions occur and affect your solution so the particles appear to have a physical meaning)?

We believe that yes they do. They can be seen as an unresolved collection of floes scattered within the smoothing length that can compact, ridge over one another, break, etc. However, since particles are points in space they cannot get in contact with one another even when their concentration is 1. Therefore, we suggest the addition of a short length contact force to simulate the collision of particles, but this is beyond the scope of our study.

The following sentence is added in the revised manuscript at l.148:  "*Since the particle density ρ is independent of the concentration, the particle concentration Ap is a quantity that measures the compactness of the floes at the particle location but does not relate to the amount of ice carried by a particle. Overall, a particle can be seen as an unresolved collection of floes scattered within the support domain A that can compact, ridge, break and drift apart. Consequently, the concentration can be interpreted as the probability of ice floes carried by a particle to come in "contact" — because particles are points in space, they never touch each other and are repulsed according to the ice strength — with ice floes of another particle.*"

Overall, do the authors consider their technique to be closer to continuum model or particle-based model?  In Section 3.2 you show that your model follows a continuum solution. While this is what you appear to be aiming for, the example raises a question for the need of the approach presented. What is the advantage of using SPH in this case (or in other

examples)? The authors should include a paragraph on this in the discussion; please emphasize what is gained by using SPH.

We consider the technique closer to the continuum model since it is based on the same sea ice dynamic equations. In its current state, the model reproduces very similar behavior as the conventional FDM formulation and doesn't bring much advantages. However, we believe that SPH enables the possibility to describe sea ice as a continuum at large scale with what is already known from the current continuum models (when there are a lot of particles) and explore some new avenues at small scales, where the continuity approximation is questionable. Indeed, the SPH discrete representation of the continuum with particles enables pairwise interactions like contact force and the conservation or transport of individual properties like angular momentum. Those further investigations inspired from DEM models, which cannot be achieved in classical continuum descriptions, are beyond the scope of the current study.

The second last paragraph has been modified to emphasize the current stage and the advantages of the method in the revised manuscript (l.471): "*In its current state, the model reproduces very similar behaviour as other FDM continuum models and does not constitute a large improvement. Nevertheless, we believe that SPH enables the possibility to describe sea ice as a continuum at large scale using what is already known from continuum models and explore some new avenues at small scales, where the continuity approximation is questionable. Indeed, SPH has interesting properties … SPH can fracture and transitions from the continuum to fragments seamlessly since it is not restricted on a grid which also has the advantage of enabling smoother ice edge shapes. The ability of the method to move around particles has the interesting property to concentrate them in converging motion increasing the resolution of the model in regions under high stress activity and to scatter them in diverging motion which decreases the resolution in low ice concentration area. This property should result in higher accuracy than typical continuum models…*"

It would be beneficial for the reader if you would include information on time step lengths and simulation times into your paper so that the reader can estimate how efficient the suggested approach is.

Agreed, at l.168 we added the information on time step length " … *time step (~ 10^-4 to 10^-2 seconds for particles of radius between 1 and 10 kilometers )* …" .

We also added a simulation time example at l.176, which now reads : "... *OpenMP. Note that in its current state, the model is rudimentary parallelized and a single time step for 40000 particles is of the order of tenths of a second. This could be greatly improved by taking advantage of CPU clusters (Yang et al., 2020) and GPUs (Xia and Liang, 2016).* ".

More detailed comments:

L37-38: If ice is thought to behave like a granular material, then is there a reason to believe the emergent properties of sea ice would not depend on floe size? Please comment on this in the text.

In the continuum formulation, the medium is considered as a whole and the ice strength at a given point only depends on the concentration and the thickness. Consequently, even if we can resolve floes of various sizes on a really fine grid, the medium property doesn't change.

We reformulated the following sentence to add some details at l. 43 : "*In practice, the emergent properties of a granular medium still depend on the assumed floe size and the nature of collisions in contrast with the continuous numerical methods which can only account for floe size in the formulation of the constitutive laws.* "

EQ17 & 18: Maybe it is mentioned somewhere in the paper, but is it common to define ice thickness and concentration as independent parameters? Maybe this is a misunderstanding by the reviewer, but at a given point in your simulation domain these two parameters cannot be totally independent, but, for example, A=0 should imply h=0. Could the authors comment on this shortly in the paper?

The ice thickness (h) and concentration (A) are independent prognostic variables in a simple two category Hibler-type model (ice or open water) and in multi-ice thickness category as well. In practice, during melt conditions, h reaches zero first, but is capped at a very small value to avoid a singularity in the A evolution equation that depends on 1/h (see for instance Hibler 1979). When h is capped to a small value, A is typically much smaller than 15%, which is considered outside of the "ice edge" in a continuum model. A continuous solution where A asymptotes to 0 and 1 (without capping for A=0 or A=1) was introduced by Gray, J. & Morland, L.W. (1994) for a more mathematically correct treatment of the mass equation, but this does not have an impact on the ice simulation.

This was clarified at line 107 of the revised manuscript with the following: "*Note that the thickness and concentration are independent prognostic variables in an Hibler-type model which can create a singularity when thickness reaches zero. To avoid this behaviour and for a more mathematically correct treatment of the mass equation, Gray and Morland (1994) introduced a continuous solution where the concentration asymptotes to 0 and 1. However, Hibler's formulation does not have an impact in our test case simulations since there is no melting and the particle thickness or concentration stay far from 0.*"

L375-379: You mention particle size does not affect jamming in your simulations. This is not what one would expect for a granular media. Does this suggest that your approach is not capable to fully represent granular behavior of an ice field (if such exists)? Do the particles have a physical meaning in your simulations? Please elaborate.

From our understanding, this means that the SPH approach is much closer to the other continuous method than to the discrete ones. If the constitutive laws don't take into account particle size and there are no contact forces added between particles then SPH doesn't incorporate the granularity by default. It only makes it easier to incorporate than for continuous methods. The physical meaning has been answered previously in the comment to answer point (5) above.

We reformulated the following sentence at l.376 to add some details: " … *no ice arch formation for floe sizes ranging from approximately one quarter to one sixteenth of the strait width. In the present model, the constitutive laws prescribe the repulsion of the particles with one another according to the ice strength, which is a function dependent on the ice concentration and mean*

*thickness, not on the particle size. We conclude that to enforce granularity within the SPH framework, the constitutive laws need to be adapted to account for contact force and particle size which could then reproduce similar behaviour as observed in DEM. However …* "

L381-384 (also FIG 8): (1) Are the "tree-like" peak stress values in your simulations transient (you use word oscillating stresses) or have your reached somewhat of a steady state in your simulations? If latter, is figure 8 just showing stress waves bouncing around the ice field in your simulation domain, which does not seem physically correct? Please clarify. If this reviewer is correct, the actual arches in your simulations appear to be limited into one close by the outlet. Please comment if you would to see more arches within the deforming ice in full-scale or if you would use DEM?

The "tree-like" peak stresses appear during transient and during the "steady state". However, note that the particles never stop moving even in a steady state because the material becomes viscous. Viscous deformation can also lead to oscillation and large stress and they are physically correct in the sense that in the SPH we approximate sea ice as a viscous-plastic material. However, it is known that SPH can have spurious behavior when the stress is solved at the same location as the particle which can be avoided if necessary (Chalk, C. Stress-Particle Smoothed Particle Hydrodynamics: An application to the failure and post-failure behavior of slopes, 2020 ).

This has been clarified in section 3.3, I., of the revised manuscript: " *From our experiments, the "tree-like" peak stresses appear during transient and at the steady state. However, the particles never stop moving even in a steady state because the material becomes viscous. Viscous deformation can lead to oscillation and large stress in the material so we hypothesized …*".

And in at l.394 : "*However, a deeper investigation is required to ensure that what is observed is physical. It is known that SPH can have spurious behaviour in some cases when the stress is solved at the same location as the particle centre (as done here) . This can be avoided using stress particles (see Chalk et al., 2020, for details).*"

Indeed, in our simulations the arch only forms close to the outlet. We know that the number of arches will increase within the deforming ice with higher resolution and they would also change location with more complex domain geometry and by changing the boundary condition to be no-slip (they are free-slip currently).

We reformulated l.407 to the following:"*... form an arch. Note that in our simulations the only one arch forms and is close to the outlet. We know that the number of arches would increase if the model is run at higher resolution and they would also change location with more complex domain geometry or by changing the boundary condition to no-slip. Overall, this… *"

L433: You again mention stress networks. Your approach adapts features from continuum models, which cannot present stress networks. Please elaborate clearly in the manuscript if you think your approach can present them reliably or not—and if yes, why does it do so if the underlying rheological model cannot present them.

Even though the approach is based on the continuum models which cannot represent stress networks, we believe that they can be observed with the SPH method because particles interact in a pairwise fashion according to their relative distance and they can move around according to

stress. This can create less dense ice areas within the medium which can lead to stress networks. However, we don't know if those are reliable since we only compared them qualitatively. More tests should be done in future work.

We reformulated to the following at line 391: " *Despite the fact that the model solve the same continuum equations as other FDM models, we believe that stress networks can be observed with the SPH method because particles interact in a pairwise fashion according to their relative distance and they can move under the action of wind/ocean forcing and internal sea ice stresses. This can create less dense ice areas within the medium which can lead to stress peaks and lows.* "
* * *
REFERENCES:

Hibler, W. D.: A Dynamic Thermodynamic Sea Ice Model,
Journal of Physical Oceanography, 9, 815–846,
https://doi.org/10.1175/1520-0485(1979)009<0815:adtsim>2.0.co;2, 1979.

Gray, J. and Morland, L.: A Two-Dimensional Model for the Dynamics of Sea Ice,
Philosophical Transactions of The Royal Society B:Biological Sciences, 347, 219–290,
https://doi.org/10.1098/rsta.1994.0045, 1994.

Chalk, C., Pastor, M., Peakall, J., Borman, D., Sleigh, P., Murphy, W., and Fuentes, R.: Stress-Particle Smoothed Particle Hydrodynamics: An application to the failure and post-failure behaviour of slopes, Computer Methods in Applied Mechanics and Engineering, 366, 113 034, https://doi.org/10.1016/j.cma.2020.113034, 2020

---

## Author Comment (AC6)

**Answers to tc-2022-163 RC3**

September 6, 2023

Note:
- The referee comments are shown in black,
- The authors answers are shown in blue,
- *Quoted texts from the revised manuscript are shown in italic and in dark red.*

\* The exact pages and line numbers in our responses are subjected to change as the revised manuscript is being prepared.
* * *
Review #3

First of all I want to thank you for submitting a well edited and relatively easy paper to read. I am happy to see the new methodology applied to the viscous-plastic model and believe the information presented will be of use to others contemplating using such a model for modeling situations where discontinuities exist in the ice pack.

Overall I think this is a useful paper and the new method to solve the model is well described. I particularly commend you for introducing the limitations of the method and discussing clearly where it can be used. My only concern is that you present only two idealized case studies to demonstrate the model works. The reproduction of the analytic solution for 1-D motion against a wall is a good test and a useful bench mark. Did you consider the range of test cases that you would need to do to demonstrate the model performance? An arching case is a classic example with a free ice edge that is a good test of the models ability to handle the discontinuity. Have you considered the work by Billy Ip and Hibler on the VP model representation of ice arching. The set up they use is different to yours, with a conical domain. They present the flow states involving arching with dimensionless numbers, and demonstrate the impact of yield curve shape on the flow through the channel. This might provide a framework for you to test your solution against.

References

Flato, G. M. (1993). A particle-in-cell sea-ice model. Atmosphere-Ocean, 31(3), 339-358.

Hibler, W. D., Hutchings, J. K., & Ip, C. F. (2006). Sea-ice arching and multiple flow states of Arctic pack ice. Annals of Glaciology, 44, 339-344.

Ip, C. F. (1993). Numerical investigation of different rheologies on sea-ice dynamics. Dartmouth College.

We have not considered the range of test cases to demonstrate the model performance yet. At this point, we mostly want to present a proof of concept showing that the SPH method can be adapted for sea ice applications, but there is still work to do before it can be applied to different model domain and forcing. The comparison of the SPH framework with standard approaches for ice arching experiment (as done in Ip and Hibler) will be addressed in future work.

Minor Suggestions

In the introduction you jump in sentence 2 stating general sea ice model architecture to the constitutive relation. For readers who are new to modeling it might help to include the information that this constitutive relation is one of the terms in the momentum balance, and how it is the continuity and momentum equations that are discretized. This is very basic, but helps guide new readers.

Agreed.

The second sentence in the introduction, l. 16, has been replaced by: " …*climate projection. Generally, numerical models used for geophysical sea-ice simulations and projections are based on a system of differential equations assuming a continuum. The equations that predict the sea ice dynamics are a combination of the momentum equations, which describe the drift of sea ice under external and internal forces, and the continuity equations which ensure mass conservation. The external forces include everything that creates a stress on sea ice and the internal forces simulate the response of the material to the external stress. The internal forces are based on various constitutive relations which can differ between models. The more commonly used constitutive laws are the standard Viscous-Plastic model (Hibler, 1979) or modifications thereof (e.g., Elastic-Viscous-Plastic or EVP and Elastic-Plastic-Anisotropic or EPA; Hunke and Dukowicz, 1997; Tsamados et al., 2013). They are typically discretized on an Eulerian mesh using finite-difference method (FDM). FDM is the simplest…*"

line 35: increase -> increased

Done.

line 45: Have you considered the work of Greg Flato under Bill Hibler? He presented a semi-lagrangian approach for solving the VP model. In his manuscript there is an example of a test case that could provide insight if used with your method. This is a free sea ice edge with a vortex forcing applied over it. In a sense you can think of this as an idealized ocean eddy at the ice edge. It was a good test case for showing how Flato's method reduced diffusion at the edge that was apparent in Hibler's solution.

We have not considered the test case of an enclosed vortex in our study because, at the moment, the boundary treatment of the SPH model is not physical and only supports free-slip conditions. The suggested work of Greg Flato would be an excellent benchmark problem to add once the model is more mature.

This has been clarified in the conclusion at l. 491: " *For future work, before exploring new features enabled by the numerical framework, a more physical treatment of the boundary conditions should be investigated — e.g., using the immersed boundary method (Tu et al., 2018) with a fixed grid for the boundary and an interpolation scheme to apply force on the particle to simulate the grounding of sea-ice near the coast. Once the boundary treatment is physically sound and enables no-slip condition, the model will be tested against other*

*benchmark problems like the ice edge modification and LKFS formation under vortex forcing (Flato, 1993; Mehlmann et al., 2021), the stress field under specific wind and water drag in an enclosed domain (Hunke, 2001; Danilov et al., 2015), or the sea ice arch problem described in n Hibler et al. (2006) to further understand and compare the effect of the SPH method. Also…"*

line 69: throughout -> through

Done.

Please check that algorithm 1 (and the tables/figures) are all referenced in the text and in order. I note that figures 8 and 9 are referenced in the text out of order.

Hopefully, everything is in order now.

We also added the following sentence at l.170 to better incorporate the algorithm in the text: "*This makes the parallelization of particle interactions algorithm mandatory for any practical applications. For further details on the computation tasks required by our application of the SPH method see Algorithm 1 below.*"

equations 33 and 35: The O(xx) terms are not explained in the text.

We clarified the terms at l.163 with the new sentence: "*In the above equations, $O(\Delta t^2)$ and $O(\Delta t^3)$ represent second-order or third-order terms and higher, which are ignored by the scheme. *"

line 203: Should there be a space between Kappa and l? Or is l a subscript? Also, please check that you are not referring to two different variables with the same variable name. Kappa is used again in equation 41, with subscript n. Do you need to use the same symbol for these two different variables?

A multiplication operator has been added between kappa and l for readability. The constant name in equation 41 has been changed to mu to avoid the possible confusion highlighted by the reviewer.

line 358: grammar is off in this sentence. I think it should be "The water drag also causes a longer  time to reach steady state, since the ice drift speed is slowed."

Agreed. The sentence at l.354 is now: *"The water drag also increases the time needed to reach steady state, since the ice drift speed is slowed."*

line 368:"than what is common" remove what.

We changed the sentence at l.363 to : *"We use a weaker wind than commonly used in Nares Strait ice arches simulations …"*

equation A17 and A18: Why use a number 1 here in place of a variable symbol?

We felt it makes it easier for the reader to associate the equation A19 this way. But as suggested, we now use the variable symbol "a".

---

## Author Response (AR2)

**Answers to tc-2022-163 RC1**

October 17, 2023

**Note:**

- The referee comments are shown in black,
- The authors answers are shown in blue,
- Quoted texts from the revised manuscript are shown in italic and in dark red.

\* The exact pages and line numbers in our responses are subjected to change as the revised manuscript is being prepared.

**Review #1**

Title: Smoothed Particle Hydrodynamics Implementation of the Standard Viscous-Plastic Sea-Ice Model and Validation in Simple Idealized Experiments

Authors: Oreste Marquis, Bruno Tremblay, Jean-François Lemieux Mohammed Islam The manuscript presents a realization of the viscous-plastic rheology with an elliptical yield curve and normal flow rule in context of the Smoothed Particle Hydrodynamics meshfree method. The authors describe the basics of the SPH method and derive the formulation of sea ice dynamics within this concept. The SPH method is evaluated on 3 simple test cases. Before reading the manuscript, I did not know the SPH method, but I am very familiar with sea ice dynamics, especially with the viscous-plastic model. I need to say that not knowing the SPH method it was hard to follow the argumentation in the paper (quite technical). Based on the presented manual I could not understand the theoretical derivation of the wave speed in Section 3.1. Looking at the numerical results it is not clear to me if the SPH approach can capture simple sea ice drift. I suggest that the authors simulate standard idealized benchmark tests of the viscous-plastic model to demonstrate that the implementation is correct and simple large-scale drift can be simulated (see main comment). I strongly recommend that the paper is reviewed by another person with a strong background on the SPH method.

**Main comment:**

I would expect that the SPH realization roughly reflects the drift of the VP model. Therefore, I would first show that simple idealized drift can be reproduced to demonstrate that the implementation is correct and the model does with one expects. In this sense the arch experiment is a bit unfortunate as the SPH method behaves different than the VP model. I suggest to solve the benchmark problem of Hunke 2001, which has been solved by Danilov et al. 2015 with removed islands. Another candidate for an evaluation would be the benchmark problem of Mehlmann et al. 2021, where the viscous-plastic model has been solved by several institutes. The formation of LKFs has been studied in this paper. It would be of interest to see if your model captures the large-scale drift and produce LKFs, which are large scale features (as the ice arches) that are coming from a small scale.

The benchmark problems suggested are interesting and would be good tests in future versions of the model. Currently, however, the no slip boundary condition is not implemented in the model; the physical representation of the boundaries in the SPH framework is challenging and could be a subject of a separate article. In its present form, a normal repulsive force on the ice is applied at the boundary and consequently, only a free slip boundary condition is permitted.

To address this comment, we will instead show the results from the 1D-SIM McGill, a standard viscous-plastic model, in the same ridging experiment and compare with that of the SPH (see Figure 5 of the revised manuscript or below). The SPH and the standard VP model are in agreement. Note that we only show at the steady state because both simulations behave differently in the transient state since we needed to reinsert the water drag to avoid convergence problems with 1D-SIM.

Further comments:

I. 20 I think the way that the sentence is phrased is not correct. Hunke does not use a classical FDM. A sub-grid discretization is applied for the approximation of the stresses. I would rephrase the sentence to: Traditionally finite difference methods haven been applied to solve the VP model.

This sentence is revised as suggested by the reviewer. The new sentence at I.22 now reads : " The more commonly used constitutive laws are the standard Viscous-Plastic model (Hibler, 1979) or modifications thereof (e.g., Elastic-Viscous-Plastic or EVP and Elastic-PlasticAnisotropic or EPA; Hunke and Dukowicz, 1997; Tsamados et al., 2013). They are typically discretized on an Eulerian mesh using finite-difference method (FDM)."

I. 116 Consistency to what?

We refer to the consistency of a discretization of a PDE. This is clarified in the revised manuscript.

The new sentence at I.540 reads: "*Finally, to ensure the consistency of the discretization of PDEs (as defined in Belytschko 1998) of the SPH method...*"

I. 188 Why do I need the information on the time step limitation? Can you please add a sentence that explains where this information is used in the ongoing analysis?

The time step limitation is used to set the time step for the SPH model. It can be calculated from a characteristic ice velocity and the radius of influence of the kernel in the model. This was clarified in the revised manuscript.

The new sentence at 1.167 now read: " The stability criterion imposes a strict limitation on the time step ( $\sim 10-4$  to 10-2 seconds for particles of radius between 1 and 10 kilometres); this cannot be avoided using a pseudo-time step because particles in an SPH framework are irregularly placed and move within the domain at each time step. This makes the parallelization of particle interactions algorithm mandatory for any practical applications. On the positive side, the explicit time stepping also eliminate possible convergence issues of the numerical solver. A pseudo-code for the proposed algorithm is shown below (Algorithm 1)."

I. 257 Please add an equation number to Gamma. The relation is used frequently in the manuscript.

We added a reference for the equation for gamma at eq.32.

I. 266 Why does it make sense to assume that the perturbation behaves like a wave solution?

This is the standard practice when studying the numerical stability of a numerical scheme. One poses a general exponential solution (f\_hat exp(i(kx-wt))), solves for k and identifies growing (unstable) mode. This was clarified in the revised manuscript.

The new sentence at I.251 now reads: "Following Williams et al. (2017), we do a perturbation analysis on the system of equations (34 - 36) and assume a wave solution of the form  $\delta f = \hat{f} \exp(i(k\bar{x} - \omega t))$ , where *i* is the imaginary number, *k* is the wavenumber,  $\omega$  is the angular velocity and *f* is a dummy variable standing for *u*, *x* and *h*."

I. 267 'the set of equations' which set? Please add equation numbers.

Clarified as suggested by the reviewer. The sentence at I. 254 now reads: "Substituting  $\delta f$  in equations (34 - 36), the resulting set of equations in the reference 11 frame following the ice motion reduces to:"

I. (49)-(51) Please add a comment where the hat notation is coming from.

The hat notation is a standard way of expressing the constant coefficient in front of the general exponential (wave) solution.

I. 273 Please add more details. Why can the summation be written in the integral form. There is no integrant in the integral. Is the integrant 1?

The integrant is everything between the integral and the dx since everything not depending on the position has been factored out. This has been clarified in the revised manuscript.

The new sentence at I.258 now reads: "For large enough wavelengths (so that the perturbation can be resolved across multiple particles with high accuracy i.e.,  $\lambda \ge lp$  and  $N \to \infty$ ), the summations can be approximated by integrals over the space i.e..."

I. 275 How do you get to the righthand side in equation (52). Can you please add some more steps?

As for Equ. 53, the exp() term is distributed and the integrals are the Fourier transforms of the derivative of the kernel (as stated on line 278). Note that at this point we use the W tilde to represent the Fourier transform of the kernel in opposition with W which represents the kernel. This has been clarified by adding steps in the derivation in the revised manuscript on line 264.

I. 279 It is unclear to me how the wave speed is derived by looking add equations (49)-(53). I stopped reading 3.1 at this line.

Equations (49 - 52 - 53) represent 49, 52 and 53 only without 50 and 51. The equation reference has been revised for clarity and, at I.266, we added the sentence: "*Finally, eqs. (37, 40 - 41) represents a system of three equations for three unknowns (* $^x$ ,  $^u$ ,  $^h$ ) that we solve by substitution. This leads to the following form for the phase speed..."

I. 338. There are several definitions of MIZ. How do you define MIZ in your setup?

We define the MIZ as the area where the sea ice concentration ranges between 0.15 and 0.8. This was clarified in the revised manuscript at I.333 : "… in the marginal ice zone (MIZ), which we define as the area where the sea ice concentration ranges between 0.15 and 0.85 and where low ridging by ice collision occurs…"

I 338. How do A and h vary in time? Based on eq. (28) and (29)? Please add some information here.

The prognostic variables h and A vary based on the dynamic processes only (i.e. divergence of div (u h)); thermodynamic processes are not considered. We added the following sentence a l. 331 : This ensures that both h and A covary in time such that h/A remains constant — note that, A and h follow the same continuity equations (15,16), or (4,5) when omitting the SPH approximations, and therefore should vary identically in time until A reaches 1 - ..."

I. 351 I think to state that numerical convergence is observed, you need to ensure that even with longer simulations no overshoots occur in fig 5 (a). Is the solution with 200[h] still approached?

Lower resolution simulations were run for a much longer time (~200 hr) and were stable. This was clarified in the revised manuscript. The new sentence added at I.319 now reads: "Results show that the simulated thickness field converges to the analytical solution (within an error of  $\approx$  1%) after  $\approx$  5 days with a slope of 1.33 × 10–3 [m · km–1], compared with 1.34 × 10–3 [m · km–1] for the theory — lower resolution simulations were run for a longer time and also converged to this stable state (results not shown)."

I.372 Why would you expect a similar sensitivity of the DEM and SPH approach? They are based on different rheologies.

The rheology in the DEM emerges from the assumptions made about the shear and normal forces between two colliding boundaries and are usually based on Mohr-Coulomb static friction law. This is different from that of Hibler's standard VP model), as stated by the reviewer. Both DEM and SPH are Lagrangian in nature, and for this reason we expect both to behave in the same manner in some circumstances. Results from Li et al., Herman, 2016; and Damsgaard et al., 2018 in similar idealized domain show similar behavior confirming this.

We have made the sentence softer in the revised manuscript. The new sentence at I.366 now reads: "We suspect that the SPH and DEM frameworks have a similar behaviour in certain circumstances even though they have different (implicit) rheologies, because of their Lagrangian nature. Indeed, the interpretation of the numerical representation of a particle in SPH as a collection of ice floes is also present in DEM (Li et al., 2014) and the two numerical frameworks compute their quantities with one-to-one interactions. Consequently, we first test whether the SPH approach has the same sensitivity to the relative size of particles with respect to the channel width as in DEM (Damsgaard et al., 2018). Results..."

**Answers to tc-2022-163 RC2**

October 17, 2023

**Note:**

- The referee comments are shown in black,
- The authors answers are shown in blue,
- Quoted texts from the revised manuscript are shown in italic and in dark red.

\* The exact pages and line numbers in our responses are subjected to change as the revised manuscript is being prepared.

**Review #2**

This paper describes the use of SPH for sea ice dynamics. In more detail, it implements VP rheology with elliptical yield curve into a SPH model. This is an interesting exercise and could lead to further work on using SPH on ice dynamics. The paper is worth publishing after some modifications. The comments by this reviewer are, mainly, related to the usefulness and applicability of the method: What is gained by using SPH when compared to FDM or DEM?

**On general level:**

Paper is very technical and it not easy to follow without a background in SPH. Is there a way to make it easier to read? Considering the readership of TC, effort to do this might increase the number of readers. Even if this reviewer is very familiar with numerical models, cannot go through all the equations of the paper. Authors could consider if such high level of detail needed here or could some parts rely on referencing earlier work? What is new in this description and what is from other sources?

In our opinion, section 2.1 is the only section which is purely theoretical and could be avoided by referencing. It describes rapidly the general concept of SPH and the kernel restrictions and assumptions. However, we feel that omitting it and just referring the theory to previous work for the readers would not give them the tools to understand the reformulation of differential equations of section 2.4 and the importance of the kernel. This is a key component to understand because from it we show that the modifications of the equations for the SPH framework modify the way plastic waves propagate in the medium. Consequently, to address reviewers' comments, we decided to keep section 2.1, but to move it to the Appendix. Sections 2.4 to 2.8 are important for people that want to know how the implementation was done and would like to reproduce or create their own SPH model. If the reviewer insists, more details could be added to specific sections.

The particle size in all simulations is of order of several kilometers. In addition, if the reviewer understands SPH correctly, all quantities in SPH become distributed over even larger area due to smoothing by kernel functions. Discussion on the following five issues in the paper is warranted:

(1) Is your model able to describe discontinuities in the deforming ice field with higher accuracy than typical continuum models (both in the case of opening leads and formation of ridges)?

As stated by the reviewer, the quantities are distributed over a large number of particles according to their smoothing length which also represent the effective resolution. From our dynamic formulation of the smoothing length and the ability of the SPH to move the particles around, we believe that during ridge formation (convergence) the discontinuities will have a high effective resolution because there are a lot of points to capture the ice deformations. On the other hand, in a lead opening (divergence) the edge of the discontinuity can be blurry because of the low number of points, but the shape of the opening is not restricted to a grid which has the advantage to enable smoother edge shape.

This sentence is added at I. 481 in the discussion of the revised manuscript: " ... in space. SPH can fracture and transitions from the continuum to fragments seamlessly since it is not restricted on a grid which also has the advantage of enabling ice edge shapes independent of it. The ability of SPH to move around particles has the interesting property to concentrate them in converging motion, effectively increasing the spatial resolution of the model in regions under high stress activity and to disperse particles when the flow is divergent which decreases the resolution in low ice concentration areas. This property should result in higher accuracy than typical continuum models The elastic ..."

**(2) Is the resolution of your model higher than typical continuum models?**

This depends on the computational power available and the parallelization efficiency of the code. Assuming the same resolution (smoothing lengths equal to the grid cells size) for both techniques the SPH method will adapt its resolution to improve it where the dynamics predict more ice (convergent flow group the particles) and reduce it in areas of low ice concentration (divergent flow scatter the particles). This is different from other continuum approaches where the grid is fixed. Therefore, SPH should improve the overall accuracy because we are usually interested in areas of high ice concentration which have high stress and deformation.

The following sentence is added at I.176 : "The proposed OpenMP parallelization is rudimentary and one time step in a domain with 40000 particles takes  $\approx 0.1$  second. For this reason, the model requires more computational resources for the effective resolution when compared with a continuum approach. This could be greatly improved by taking advantage of CPU clusters (Yang et al., 2020) or GPUs (Xia and Liang, 2016)."

(3) is the coarse resolution, or large particle size, due to computational burden?

Yes it is. The SPH method is explicit, which cannot take advantage of the solvers used in FDM like JNFK or Picard-SOR. The SPH explicit formulation forces a really small time step of the order of the hundredths of a second for a smoothing length of 10 km to properly resolve the plastic-wave propagation. This is around 6 orders of magnitude smaller than the time step used in FDM. The efficiency of SPH comes from its great potential in parallelization which, we believe, could compete with FDM on supercomputers for large simulations.

The following sentence is added in the discussion section 2.4 in the revised at 1.167" The stability criterion imposes a strict limitation on the time step (~ 10-4 to 10-2 seconds for particles of radius between 1 and 10 kilometres); this cannot be avoided using a pseudo-time step because particles in an SPH framework are irregularly placed and move within the domain at each time step. This makes the parallelization of particle interactions algorithm mandatory for any practical applications. On the positive side, the explicit time stepping also eliminate possible convergence issues of the numerical solver. A pseudo-code for the proposed algorithm is shown below (Algorithm 1)."

(4) Does it even make sense to decrease the particle size when VP rheology is used? At the moment, the dynamic formulation of sea ice used in the model is fairly simple and an increase in resolution (or decrease in particle size) is not really useful. The model uses the VP rheology as a test case because it is well known and makes the comparison with previous work easier, but further development using SPH should step away from it.

The following sentence is at I.72 in the revised manuscript: "In this work, we use the standard VP sea-ice model with an elliptical yield curve and normal flow rule (Hibler, 1979) as a proof-of-concept. Further development of the SPH model should consider a broader range of rheologies. We..."

(5) Does an individual particle in your simulation have physical meaning (do they, for example, describe ice floes – you do mention that particle collisions occur and affect your solution so the particles appear to have a physical meaning)?

We believe that yes they do. They can be seen as an unresolved collection of floes scattered within the smoothing length that can compact, ridge over one another, break, etc. However, since particles are points in space they cannot get in contact with one another even when their concentration is 1. Therefore, we suggest the addition of a short length contact force to simulate the collision of particles, but this is beyond the scope of our study.

The following sentence is added in the revised manuscript at 1.148: "Overall, a particle can be seen as an unresolved collection of floes scattered within the support domain A that can converge, ridge over one another, break and drift apart. Note that since the particle density pp definition is independent of Ap, the concentration can be interpreted as a quantity that measures the compactness of the sea-ice at the particle location. It describes the probability of ice floes carried by a particle, which is a point in space, to come in "contact" with ice floes of another particle and get repulsed according to the ice strength."

Overall, do the authors consider their technique to be closer to continuum model or particlebased model? In Section 3.2 you show that your model follows a continuum solution. While this is what you appear to be aiming for, the example raises a question for the need of the approach presented. What is the advantage of using SPH in this case (or in other examples)? The authors should include a paragraph on this in the discussion; please emphasize what is gained by using SPH.

We consider the technique closer to the continuum model since it is based on the same sea ice dynamic equations. In its current state, the model reproduces very similar behavior as the conventional FDM formulation and doesn't bring much advantages. However, we believe that SPH enables the possibility to describe sea ice as a continuum at large scale with what is already known from the current continuum models (when there are a lot of particles) and explore some new avenues at small scales, where the continuity approximation is questionable. Indeed, the SPH discrete representation of the continuum with particles enables pairwise interactions like contact force and the conservation or transport of individual properties like angular momentum. Those further investigations inspired from DEM models, which cannot be achieved in classical continuum descriptions, are beyond the scope of the current study.

The second last paragraph has been modified to emphasize the current stage and the advantages of the method in the revised manuscript (I.474): "In its current state, the model reproduces very similar behaviour as other FDM continuum models and does not constitute a large improvement. Nevertheless, we believe that SPH enables the possibility to describe sea ice as a continuum at large scale using what is already known from continuum models and explore some new avenues at small scales, where the continuity approximation is questionable. Indeed, SPH also has interesting properties... SPH can fracture and transitions from the continuum to fragments seamlessly since it is not restricted on a grid which also has the advantage of enabling ice edge shapes independent of it. The ability of SPH to move around particles has the interesting property to concentrate them in converging motion, effectively increasing the spatial resolution of the model in regions under high stress activity and to disperse particles when the flow is divergent which decreases the resolution in low ice concentration areas. This property should result in higher accuracy than typical continuum models. The elastic..."

It would be beneficial for the reader if you would include information on time step lengths and simulation times into your paper so that the reader can estimate how efficient the suggested approach is.

Agreed, at I.167 we added the information on time step length " ... time step (~ 10^-4 to 10^-2 seconds for particles of radius between 1 and 10 kilometers ) ... ".

We also added a simulation time example at I.175, which now reads : "... OpenMPThe proposed OpenMP parallelization is rudimentary and one time step in a domain with 40000 particles takes  $\approx 0.1$  second. For this reason, the model requires more computational resources for the effective resolution when compared with a continuum approach. This could be greatly improved by taking advantage of CPU clusters (Yang et al., 2020) or GPUs (Xia and Liang, 2016). ".

More detailed comments:

L37-38: If ice is thought to behave like a granular material, then is there a reason to believe the emergent properties of sea ice would not depend on floe size? Please comment on this in the text.

In the continuum formulation, the medium is considered as a whole and the ice strength at a given point only depends on the concentration and the thickness. Consequently, even if we can resolve floes of various sizes on a really fine grid, the medium property doesn't change.

We reformulated the following sentence to add some details at I. 42 : " In practice, the emergent properties of a granular medium still depend on the assumed floe size and the nature of collisions in contrast with the continuous numerical methods which can which indirectly account for floe interactions through the formulation of a constitutive law."

EQ17 & 18: Maybe it is mentioned somewhere in the paper, but is it common to define ice thickness and concentration as independent parameters? Maybe this is a misunderstanding by the reviewer, but at a given point in your simulation domain these two parameters cannot be totally independent, but, for example, A=0 should imply h=0. Could the authors comment on this shortly in the paper?

The ice thickness (h) and concentration (A) are independent prognostic variables in a simple two category Hibler-type model (ice or open water) and in multi-ice thickness category as well. In practice, during melt conditions, h reaches zero first, but is capped at a very small value to avoid a singularity in the A evolution equation that depends on 1/h (see for instance Hibler 1979). When h is capped to a small value, A is typically much smaller than 15%, which is considered outside of the "ice edge" in a continuum model. A continuous solution where A asymptotes to 0 and 1 (without capping for A=0 or A=1) was introduced by Gray, J. & Morland, L.W. (1994) for a more mathematically correct treatment of the mass equation, but this does not have an impact on the ice simulation.

This was clarified at line 108 of the revised manuscript with the following: "Note that the thickness and concentration are independent prognostic variables in a two-category model (Hibler, 1979), resulting in a singularity when thickness is reaches zero. To avoid singularity and for a more mathematically correct treatment of the mass equation, Gray and Morland (1994) introduced a continuous solution where the concentration asymptotes to zero and one. In the following, we ignore melting and consider cases where only convergent motion is present only, and the use of a two-category model does not have an impact on the simulated results."

L375-379: You mention particle size does not affect jamming in your simulations. This is not what one would expect for a granular media. Does this suggest that your approach is not capable to fully represent granular behavior of an ice field (if such exists)? Do the particles have a physical meaning in your simulations? Please elaborate.

From our understanding, this means that the SPH approach is much closer to the other continuous method than to the discrete ones. If the constitutive laws don't take into account particle size and there are no contact forces added between particles then SPH doesn`t incorporate the granularity by default. It only makes it easier to incorporate than for continuous methods. The physical meaning has been answered previously in the comment to answer point (5) above.

We reformulated the following sentence at I.376 to add some details: "... no ice arch formation for floe sizes ranging from approximately one quarter to one sixteenth of the strait width. We explain this difference between SPH and DEM from the continuum description of the ice dynamics equation. In the present model, the constitutive laws prescribe the repulsion of the

particles with one another according to the ice strength, which is a function dependent on the ice concentration and mean thickness, not on the particle size. We conclude that to enforce granularity within the SPH framework, the constitutive laws would need to be adapted to account for contact force and particle size which could then reproduce similar behaviour as observed in DEM. However ... "

L381-384 (also FIG 8): (1) Are the "tree-like" peak stress values in your simulations transient (you use word oscillating stresses) or have your reached somewhat of a steady state in your simulations? If latter, is figure 8 just showing stress waves bouncing around the ice field in your simulation domain, which does not seem physically correct? Please clarify. If this reviewer is correct, the actual arches in your simulations appear to be limited into one close by the outlet. Please comment if you would to see more arches within the deforming ice in full-scale or if you would use DEM?

The "tree-like" peak stresses appear during transient and during the "steady state". However, note that the particles never stop moving even in a steady state because the material becomes viscous. Viscous deformation can also lead to oscillation and large stress and they are physically correct in the sense that in the SPH we approximate sea ice as a viscous-plastic material. However, it is known that SPH can have spurious behavior when the stress is solved at the same location as the particle which can be avoided if necessary (Chalk, C. Stress-Particle Smoothed Particle Hydrodynamics: An application to the failure and post-failure behavior of slopes, 2020 ).

This has been clarified in section 3.3, I.385, of the revised manuscript: " From our experiments, the "tree-like" peak stresses appear during transient and at steady-state. However, the particles never stop moving even in a steady-state because viscous deformations are always present. We hypothesized that stress patterns are associated with over-damped elastic waves associated with small movement (but large internal stresses) of the particle in the viscous regime".

And in at 1.395 : "It is known that SPH can have spurious behaviour in some cases when the stress is solved at the same location as the particle centre (as done here). This can be avoided using stress particles (see Chalk et al., 2020, for details). More investigations are required to test whether this behaviour is physical. This is left for future work."

Indeed, in our simulations the arch only forms close to the outlet. We know that the number of arches will increase within the deforming ice with higher resolution and they would also change location with more complex domain geometry and by changing the boundary condition to be no-slip (they are free-slip currently).

We reformulated I.409 to the following:" Note that in the SPH simulations, only one arch forms close to the outlet. Presumably, the number of arches would increase and location would change if the model was run at higher resolution, with different boundary conditions or in a non-idealized domain geometry."

L433: You again mention stress networks. Your approach adapts features from continuum models, which cannot present stress networks. Please elaborate clearly in the manuscript if you think your approach can present them reliably or not—and if yes, why does it do so if the underlying rheological model cannot present them.

Even though the approach is based on the continuum models which cannot represent stress networks, we believe that they can be observed with the SPH method because particles interact in a pairwise fashion according to their relative distance and they can move around according to stress. This can create less dense ice areas within the medium which can lead to stress networks. However, we don't know if those are reliable since we only compared them qualitatively. More tests should be done in future work.

We reformulated to the following at line 393: " Despite the fact that the model solves the same continuum equations as other FDM models, we believe that stress networks can be observed with the SPH method because particles interact in a pairwise fashion according to their relative distance. This can create less dense ice areas within the medium which can lead to oscillations in the stress field."

**REFERENCES**:**

Hibler, W. D.: A Dynamic Thermodynamic Sea Ice Model, Journal of Physical Oceanography, 9, 815–846, https://doi.org/10.1175/1520-0485(1979)009<0815:adtsim>2.0.co;2, 1979.

Gray, J. and Morland, L.: A Two-Dimensional Model for the Dynamics of Sea Ice, Philosophical Transactions of The Royal Society B:Biological Sciences, 347, 219–290, https://doi.org/10.1098/rsta.1994.0045, 1994.

Chalk, C., Pastor, M., Peakall, J., Borman, D., Sleigh, P., Murphy, W., and Fuentes, R.: Stress-Particle Smoothed Particle Hydrodynamics: An application to the failure and post-failure behaviour of slopes, Computer Methods in Applied Mechanics and Engineering, 366, 113 034, https://doi.org/10.1016/j.cma.2020.113034, 2020

**Answers to tc-2022-163 RC3**

October 17, 2023

**Note:**

- The referee comments are shown in black,
- The authors answers are shown in blue,
- Quoted texts from the revised manuscript are shown in italic and in dark red.

\* The exact pages and line numbers in our responses are subjected to change as the revised manuscript is being prepared.

**Review #3**

First of all I want to thank you for submitting a well edited and relatively easy paper to read. I am happy to see the new methodology applied to the viscous-plastic model and believe the information presented will be of use to others contemplating using such a model for modeling situations where discontinuities exist in the ice pack.

Overall I think this is a useful paper and the new method to solve the model is well described. I particularly commend you for introducing the limitations of the method and discussing clearly where it can be used. My only concern is that you present only two idealized case studies to demonstrate the model works. The reproduction of the analytic solution for 1-D motion against a wall is a good test and a useful bench mark. Did you consider the range of test cases that you would need to do to demonstrate the model performance? An arching case is a classic example with a free ice edge that is a good test of the models ability to handle the discontinuity. Have you considered the work by Billy Ip and Hibler on the VP model representation of ice arching. The set up they use is different to yours, with a conical domain. They present the flow states involving arching with dimensionless numbers, and demonstrate the impact of yield curve shape on the flow through the channel. This might provide a framework for you to test your solution against.

**References**

Flato, G. M. (1993). A particle-in-cell sea-ice model. Atmosphere-Ocean, 31(3), 339-358.

Hibler, W. D., Hutchings, J. K., & Ip, C. F. (2006). Sea-ice arching and multiple flow states of Arctic pack ice. Annals of Glaciology, 44, 339-344.

Ip, C. F. (1993). Numerical investigation of different rheologies on sea-ice dynamics. Dartmouth College.

We have not considered the range of test cases to demonstrate the model performance yet. At this point, we mostly want to present a proof of concept showing that the SPH method can be adapted for sea ice applications, but there is still work to do before it can be applied to different model domain and forcing. The comparison of the SPH framework with standard approaches for ice arching experiment (as done in Ip and Hibler) will be addressed in future work.

**Minor Suggestions**

In the introduction you jump in sentence 2 stating general sea ice model architecture to the constitutive relation. For readers who are new to modeling it might help to include the information that this constitutive relation is one of the terms in the momentum balance, and how it is the continuity and momentum equations that are discretized. This is very basic, but helps guide new readers.

**Agreed.**

The second sentence in the introduction, I. 16, has been replaced by: "...climate projection. Generally, numerical models used for geophysical sea-ice simulations and projections are based on a system of differential equations assuming a continuum. The equations that predict the sea ice dynamics are a combination of the momentum equations, which describe the drift of sea ice under external and internal forces, and the continuity equations which ensure mass conservation. The 20 external forces (per unit area) generally include surface air stress, water drag, sea surface tilt and the Coriolis effect and the internal forces are related to the ice stress term. This internal stress term is based on various constitutive relations which can differ between models. The more commonly used constitutive laws are the standard Viscous-Plastic model (Hibler, 1979) or modifications thereof (e.g., Elastic-Viscous-Plastic or EVP and Elastic-Plastic-Anisotropic or EPA; Hunke and Dukowicz, 1997; Tsamados et al., 2013). They are typically discretized on an Eulerian mesh using finite-difference method (FDM). FDM is the simplest..."

line 35: increase -> increased

**Done.**

line 45: Have you considered the work of Greg Flato under Bill Hibler? He presented a semilagrangian approach for solving the VP model. In his manuscript there is an example of a test case that could provide insight if used with your method. This is a free sea ice edge with a vortex forcing applied over it. In a sense you can think of this as an idealized ocean eddy at the ice edge. It was a good test case for showing how Flato's method reduced diffusion at the edge that was apparent in Hibler's solution.

We have not considered the test case of an enclosed vortex in our study because, at the moment, the boundary treatment of the SPH model is not physical and only supports free-slip conditions. The suggested work of Greg Flato would be an excellent benchmark problem to add once the model is more mature.

This has been clarified in the conclusion at I. 494: "For future work, and before exploring new features enabled by the SPH numerical framework, a more physical treatment 495 of the boundary conditions should be investigated to properly simulate the grounding of sea-ice near the coast enabling the no-slip conditions. Subsequently, the model could be tested against other benchmark problems in idealized domain to further understand and compare the effect of the

SPH method (Flato, 1993; Hunke, 2001; Hibler et al., 2006; Danilov et al., 2015; Mehlmann et al., 2021)."

line 69: throughout -> through

Done.

Please check that algorithm 1 (and the tables/figures) are all referenced in the text and in order. I note that figures 8 and 9 are referenced in the text out of order.

Hopefully, everything is in order now.

We also added the following sentence at I.170 to better incorporate the algorithm in the text: "A pseudo-code for the proposed algorithm is shown below (Algorithm 1)."

equations 33 and 35: The O(xx) terms are not explained in the text.

We clarified the terms at I.162 with the new sentence: "In the above equations,  $O(\Delta t \ 2)$  and  $O(\Delta t \ 3)$  represent higher-order terms, which are ignored in the proposed scheme."

line 203: Should there be a space between Kappa and I? Or is I a subscript? Also, please check that you are not referring to two different variables with the same variable name. Kappa is used again in equation 41, with subscript n. Do you need to use the same symbol for these two different variables?

A multiplication operator has been added between kappa and I for readability. The constant name in equation 41 has been changed to mu to avoid the possible confusion highlighted by the reviewer.

line 358: grammar is off in this sentence. I think it should be "The water drag also causes a longer time to reach steady state, since the ice drift speed is slowed."

Agreed. The sentence at I.354 is now: "The water drag also increases the time needed to reach steady-state, because the ice drift speed is slower."

line 368:"than what is common" remove what.

We changed the sentence at I.364 to : "We use a weaker wind than commonly used in Nares Strait ice arches simulations ..."

equation A17 and A18: Why use a number 1 here in place of a variable symbol?

We felt it makes it easier for the reader to associate the equation A19 this way. But as suggested, we now use the variable symbol "a".